# AGENTIC REINFORCEMENT LEARNING WITH IMPLICIT STEP REWARDS

**Xiaoqian Liu**[1,2,3,†]**, Ke Wang**[2]**, Yuchuan Wu**[2]**, Fei Huang**[2]**, Yongbin Li**[2]**,
Jianbin Jiao**[1] **and Junge Zhang**[3,1]*

[1]University of Chinese Academy of Sciences. `liuxiaoqian23@mails.ucas.ac.cn`
[2]Tongyi Lab. `{wk258730,shengxiu.wyc,shuide.lyb}@alibaba-inc.com`
[3]The Key Laboratory of Cognition and Decision Intelligence for Complex Systems,
Institute of Automation, Chinese Academy of Sciences. `jgzhang@nlpr.ia.ac.cn`

## ABSTRACT

Large language models (LLMs) are increasingly developed as autonomous agents using reinforcement learning (agentic RL) that reason and act in interactive environments. However, sparse and sometimes unverifiable rewards make it extremely challenging to assign credit when training LLM agents that serve as a policy. Recent work attempts to integrate process supervision into RL but suffers from biased annotation, reward hacking, high-variance from overly fine-grained rewards or failures when state overlap is rare. We therefore introduce implicit step rewards for agentic RL (**iStar**), a general credit-assignment strategy that integrates seamlessly with standard RL algorithms without relying on additional rollouts or explicit step labels. Particularly, we alternatively optimize an implicit process reward model (PRM) with the policy model to generate step rewards for each action via a multi-turn DPO objective. Theoretical analysis shows that this learning objective produces a step-wise reward function learned from trajectory preferences. Then the implicit step rewards are used to compute step-level advantages, which are combined with trajectory (or episode)-level advantages for policy updates, creating a self-reinforcing training loop. We evaluate our method on three challenging agent benchmarks, including WebShop and VisualSokoban, as well as open-ended social interactions with unverifiable rewards in SOTOPIA. Crucially, our method shows superior performance over frontier LLMs and strong RL baselines across domains, achieving state-of-the-art results with higher sample-efficiency and training stability. Further analysis also demonstrates efficient exploration by **iStar** with increased rewards in both step- and episode-level while maintaining fewer steps to achieve task success.

## 1 INTRODUCTION

LLMs are rapidly evolving from passive generators into autonomous agents that can reason, act, and adapt strategies over long horizons, including search agents (Jin et al., 2025; OpenAI, 2025a), mobile and web navigators (Furuta et al., 2024; Bai et al., 2024), software engineering assistants (Yang et al., 2025; Wei et al., 2025a), and social or embodied intelligence (Liu et al., 2025; Lu et al., 2025). Unlike conventional RL for LLM post-training in static, single-turn tasks (Ouyang et al., 2022; Shao et al., 2024; Wang et al., 2026), training LLM agents in interactive environments faces particular challenges: (1) rewards are typically sparse and delayed, complicating credit assignment to intermediate actions; (2) trajectories are long and non-Markovian in token level, with each step consisting of a chain-of-thought (CoT) (Wei et al., 2022) and an executable action, inflating variance when credit is pushed to individual tokens; and (3) environments and counterparts are non-stationary, open-ended and often come with unverifiable rewards (e.g, dialogues). Consequently, trajectory-level optimization with a single outcome reward (Wang et al., 2025; Wei et al., 2025b) suffers from credit assignment, yielding high-variance policy learning, brittle exploration, and limited gains on agent tasks.

---

*Corresponding author
†Work done during an internship at Tongyi Lab.

Recent work has attempted to solve these problems particularly through process supervision in RL. For example, Zeng et al. (2025); Zou et al. (2025); Zhang et al. (2025b) provide denser feedback at intermediate steps but require handcrafted step labels that are costly, biased, and vulnerable to reward hacking. Generative reward models (GRMs) (e.g, LLM-as-judge) (Liu et al., 2025; Zha et al., 2025) that predict criticality or correctness for each step reduce annotation overhead but can be noisy and inconsistent across domains. Implicit PRMs (Yuan et al., 2025; Cui et al., 2025) help in single-turn tasks, but the token-level process rewards tend to be overly fine-grained in agent training, amplifying variance and destabilizing training as trajectories grow. Other approaches (Feng et al., 2025; Choudhury, 2025) compute step-level advantages by grouping identical states, an assumption that fails in open-ended language environments where state overlap is rare. Together, these limitations raise a core question for agentic RL: **How can we design a credit-assignment strategy that is label-efficient and stable, scales to multi-turn interactions, and remains robust and generalizable to (un)verifiable rewards in open-ended environments?**

To address this, we propose implicit step rewards for agentic RL (**iStar**), a general credit-assignment strategy for LLM agents. **iStar** jointly trains an implicit PRM with the policy model (the LLM agent) using trajectories collected online. At each training step, the policy model generates rollouts, which are ranked by an outcome reward verifier (or model) to form positive–negative trajectory pairs. The implicit PRM is optimized via a multi-turn DPO objective on these pairs. The updated PRM then generates implicit step rewards for each action by measuring its relative preference over the previous policy snapshot. Since this reward is computed per step, it provides dense feedback to guide exploration while staying coarse enough to keep variance under control. When training the policy model, we combine two complementary advantages: an episode-level advantage from outcome rewards and a step-level advantage from the implicit step rewards, capturing both global task success and the contribution of individual actions. **iStar** is compatible with various existing RL algorithms, such as GRPO (Shao et al., 2024), RLOO (Ahmadian et al., 2024), and DAPO (Yu et al., 2025), without relying on annotated step labels or additional rollouts.

Our method addresses the limitations of prior work along several dimensions: (1) **iStar** provides step credit without annotated labels through implicit rewards derived from a multi-turn DPO objective, which is guaranteed to be a step-wise reward function learned from trajectory preferences (see Section 3.2); (2) **iStar** stabilizes training with multi-turn RL by learning implicit rewards at the step level rather than the token-level (Cui et al., 2025); (3) **iStar** only relies on trajectory preferences that can be sourced from (un)verifiable outcome rewards even in open-ended environments, enabling unified credit assignment across different domains.

Experiments on three challenging agent benchmarks show that **iStar** achieves superior performance over frontier LLMs and strong RL baselines, achieving state-of-the-art results in WebShop and VisualSokoban. In SOTOPIA, open-ended social interactions with unverifiable rewards, **iStar** increases goal completion by up to 14% in self-chat and 48% when chatting with GPT-4o (OpenAI, 2024). Our method can also be plugged into different RL algorithms to boost their performance. Further analysis shows higher sample efficiency and training stability of our method than vanilla RL and token-level PRM (Cui et al., 2025) baselines, as well as efficient exploration with increased rewards in both step- and episode-level and fewer steps to achieve a goal. Our main contributions include:

- We introduce a general credit assignment strategy for LLM agentic (multi-turn) RL that alternatively optimizes an implicit step-level PRM with the policy model, creating a self-reinforcing training loop.

- We propose a multi-turn DPO objective to optimize the implicit PRM online, which is theorectically guaranteed to be a step-wise reward function learned from trajectory preferences.

- Experiments demonstrate superior performance of our method over baselines across various environments, showing higher sample efficiency, training stability and robustness to unverifiable rewards, as well as efficient exploration during multi-turn online RL.

## 2 PRELIMINARIES

**Task formulation.** We consider the LLM agent task as a multi-step decision-making process, where the agent interacts with the environment to achieve a long-term goal through sequential decisions given a task prompt $x \in p(X)$. At each timestep $t$, the agent receives an observation $o_t$

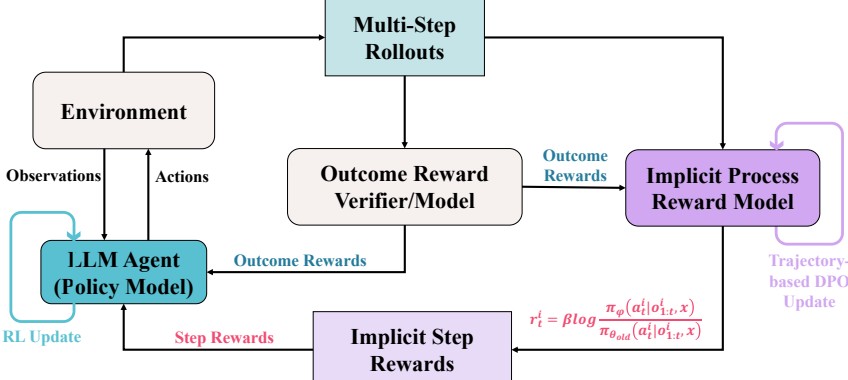

Figure 1: **Overview of iStar.** At each training step, an LLM agent interacts with the environment to generate multi-step rollouts ranked by an outcome reward verifier (or model) to construct positive-negative trajectory pairs. These pairs are used to train an implicit PRM via a multi-turn DPO objective, which generates implicit step rewards for each action produced by the agent. Finally, calculate step-level advantages using the implicit step rewards and episode-level advantages using outcome rewards to optimize the LLM agent (policy model) through RL.

from the environment and responds with a textual action $a_t \in V^L$, where $V$ denotes the token vocabulary and $L$ the maximum generation length[1]. The environment then returns a scalar reward $r_t$ and transitions to the next state. Until the last timestep $T$, the full episode consists of a trajectory $\tau = \{(o_1, a_1, r_1), ..., (o_T, a_T, r_T)\}$. However, in real-world scenarios, rewards can be sparse and delayed, such as feedback provided only at the end of a trajectory.

**RL for LLMs.** RL addresses the agent task by optimizing the LLM agent $\pi_\theta(a_t|o_{1:t}, x)$, with the objective of maximizing the expected cumulative rewards during multi-turn interactions. Policy gradient methods are usually used, such as PPO (Schulman et al., 2017), GRPO (Shao et al., 2024), RLOO (Ahmadian et al., 2024), and REINFORCE++ (Hu et al., 2025). These RL algorithms mainly differ in the manner of estimating advantages for policy updates. For example, PPO computes advantages with a learned value function using generalized advantage estimation. GRPO and RLOO are critic-free and form relative advantages within a group of $N$ responses for the same input prompt. GRPO centers (and often normalizes) each reward by the group mean, while RLOO uses a leave-one-out mean. REINFORCE++ instead uses batch-normalized rewards as the baseline reward.

**Implicit reward modeling.** Implicit rewards have shown effectiveness in reward modeling for LLM alignment by enabling models to infer rewards without explicit labels (Ethayarajh et al., 2024; Wu et al., 2025; Zhang et al., 2025a). The implicit rewards are used to evaluate the quality of model outputs, such as DPO (Rafailov et al., 2024b)). Further, Rafailov et al. (2024a) demonstrates that DPO can automatically learn a Q-function. Beyond the use as reward models or Q-functions, recent work uses implicit token-level rewards for test-time reranking (Yuan et al., 2025) or single-turn RL (Cui et al., 2025): $r_\phi(y_t) := \beta \log \frac{\pi_\phi(y_t|y_{<t})}{\pi_{\text{ref}}(y_t|y_{<t})}$, where $\pi_\phi$ represents the token-level reward model and $\pi_{\text{ref}}$ the reference model. $y_t$ denotes the $t$-th token in the response $y$.

## 3 METHOD

In this section, we first provide an overview of our method with a definition of implicit step rewards for agentic RL. We then present theoretical analysis to justify the learning objective of the implicit PRM in **iStar** produces a step-wise reward function.

---

[1]Note that we prompt the LLM agent to produce a reasoning process before executing an action, and $a_t$ refers to the whole sequence consisting of both the reasoning and the action at each timestep.

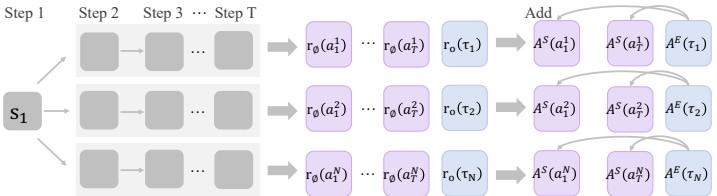

Figure 2: **The credit-assignment strategy of iStar.** In our method, episode-level advantages $A^E(\tau)$ are computed using outcome rewards $r_o(\tau)$, while step-level advantages $A^S(a)$ are calculated based on implicit step rewards $r_\phi(a)$ produced by the implicit PRM. The final advantages for policy updates is a combination of these two-level advantages.

## 3.1 OVERVIEW

In **iStar**, there is an implicit PRM optimized alternately with the policy model (LLM agent), transforming the tendency to prefer more optimal actions into dense step rewards to guide exploration and improvement of the agent. Figure 1 shows the overall training pipeline of our method. The alternating optimization between the implicit PRM and the policy model creates a self-reinforcing training loop to iteratively enhance each other. Below we first provide a definition of the implicit step rewards, and detail the training process for the implicit PRM and the policy model, respectively. Please refer to Appendix A for the detailed algorithm of **iStar**.

**Implicit step rewards.** Let $\tau = (o_1, a_1, ..., o_T, a_T)$ denote a trajectory produced by the LLM agent using policy $\pi_\theta$. For action $a_t$ in the trajectory at step $t$, its implicit step reward is defined as

$$r_\phi(o_{1:t}, a_t) = \beta \log \frac{\pi_\phi(a_t|o_{1:t}, x)}{\pi_{\theta_{\text{old}}}(a_t|o_{1:t}, x)}, \tag{1}$$

where $\pi_\phi$ represents the implicit PRM, $\pi_{\theta_{\text{old}}}$ refers to the previous snapshot of the policy $\pi_\theta$, and $\beta \in [0, 1]$ is a temperature that scales the reward. The implicit step reward measures how much more probable the current action is under the freshly learned PRM than under the old policy. Positive values indicate actions that $\pi_\phi$ believes to be responsible for recent improvements, while negative values highlight actions that should be discouraged.

**Optimizing implicit PRM via multi-turn DPO online.** For scalable online RL, we train the implicit PRM $\pi_\phi$ on positive-negative trajectory pairs sampled by the old policy model $\pi_{\theta_{\text{old}}}$ and derive a multi-turn DPO objective for online update:

$$\mathcal{J}_{\text{PRM}}(\phi) = -\mathbb{E}_{\substack{(\tau^+, \tau^-) \sim \pi_{\theta_{\text{old}}} \\ x \sim p(X)}} \left[ \log \sigma \left( \beta \log \frac{\pi_\phi(\tau^+ \mid x)}{\pi_{\theta_{\text{old}}}(\tau^+ \mid x)} - \beta \log \frac{\pi_\phi(\tau^- \mid x)}{\pi_{\theta_{\text{old}}}(\tau^- \mid x)} \right) \right], \tag{2}$$

where $\sigma$ is the logistic sigmoid, and $\pi_\phi$, $\pi_{\theta_{\text{old}}}$ as well as $\beta$ follows Eq. 1. $\pi(\tau|x)$ refers to the product of action probabilities controlled by the policy (i.e., $\prod_t \pi(a_t \mid o_{1:t}, x)$. $\tau^+$ is a positive trajectory that is preferred to the negative one $\tau^-$, both of which are labeled by an outcome reward verifier (or model) [2]. Particularly, Eq. 2 has two main differences from the standard DPO (Rafailov et al., 2024b): (1) we use the previous policy snapshot $\pi_{\theta_{\text{old}}}$ as the reference model, whose parameters alter during training, instead of the initial policy model that keeps frozen; (2) we derive the DPO objective from a multi-step Markov decision process (MDP) rather than a one-step bandit problem (Rafailov et al., 2024b). See Section 3.2 for the justification that Eq. 2 is a multi-turn version of standard DPO that equivalents to a Bradley-Terry (BT) model with a step-wise reward function.

**Policy learning with implicit step rewards.** We use GRPO (Shao et al., 2024) as an example to illustrate how to integrate our implicit step rewards into policy learning, despite that our method is compatible with a variety of RL algorithms (Ahmadian et al., 2024; Hu et al., 2025; Yu et al., 2025; Zheng et al., 2025). As shown in Figure 2, for each task prompt $x$, we sample a group of

---

[2]In our experiments on WebShop and VisualSokoban, positive trajectories are those with success rates above 0, while for SOTOPIA, positive trajectories have goal completion score above 6.

$N$ trajectories $\{\tau_1, ..., \tau_N\}$ from the old policy $\pi_{\theta_{\text{old}}}$, and obtain its corresponding outcome rewards $\{r_o(\tau_1), ..., r_o(\tau_N)\}$ through an outcome reward verifier (or model). We then compute episode-level advantages $A^E$ for the group of trajectories:

$$A^E(\tau_i) = \big(r_o(\tau_i) - mean(R_o)\big)/std(R_o), \tag{3}$$

where $R_o$ is the set of outcome rewards of $N$ trajectories $r_o(\tau_i)_{i=1}^N$. Next, we use the latest implicit PRM $\pi_\phi$ (from the previous training step) to obtain implicit step reward for each action $a_t^i$ via Eq. 1, and compute step-level advantages:

$$A^S(a_t^i) = \big(r_\phi(a_t^i) - mean(R_s)\big)/std(R_s), \tag{4}$$

where $R_s = r_\phi(a_t^i)|i, t$ is the set of all step rewards in the $N$ trajectories. Particularly, when we use a group of trajectories starting from the same initial state (task prompt), we can generate various counterfactual scenarios. This helps us calculate a more accurate and stable estimate of the state-value baselines, leading to a better measure of the advantage for $a_t^i$. On the other hand, in a single trajectory, actions happen in different states and are influenced by noise specific to the policy used by the agent, leading to advantages in high variance.

Given episode-level advantages and step-level advantages, we combine them for policy updates:

$$A(a_t^i) = A^E(\tau_i) + \alpha A^S(a_t^i), \tag{5}$$

where $\alpha$ is a hyperparameter that balances the two-level advantages. The final advantage can differentiate not only between favorable and unfavorable trajectories but also beneficial and detrimental steps within a group of trajectories from the same initial state, enabling more dense rewards for policy learning in long-horizons.

Ultimately, given the advantages $A(a_t^i)$, the policy model $\pi_\theta$ is trained using a surrogate objective that is widely used in Schulman et al. (2017); Shao et al. (2024):

$$\mathcal{J}_{\text{policy}}(\theta) = \mathbb{E}_{\substack{\{\tau_i\}_{i=1}^N \sim \pi_{\theta_{\text{old}}} \\ x \sim p(X)}} \left[ \frac{1}{NT} \sum_{i=1}^N \sum_{t=1}^{T_i} \min\big(\rho_\theta(a_t^i)A(a_t^i), \text{clip}\big(\rho_\theta(a_t^i), 1 \pm \epsilon\big) A(a_t^i)\big) \right], \tag{6}$$

where $\rho_\theta(a_t^i) = \frac{\pi_\theta(a_t^i|\mathbf{o}_t^i,\mathbf{x})}{\pi_{\theta_{\text{old}}}(a_t^i|\mathbf{o}_t^i,\mathbf{x})}$ is the importance sampling ratio at the step-level, and $\epsilon$ is a hyperparameter that controls the clipping range of the importance. Particularly, the step-level importance sampling ratio aligns well with our implicit step rewards to ensure low-variance training noise on multi-step rollouts, similar to (Zheng et al., 2025). We also choose not to apply any KL-divergence penalty to the policy model during training. In an online, agentic RL setting, successful behavior may need to deviate substantially from what a frozen language model would normally produce. Removing the KL penalty allows the policy model to explore the action space more freely, especially in regions that are crucial for solving the task. See Table 7 for performance comparison with and without the KL-divergence penalty.

**Remarks.** Alternating optimization between the implicit PRM and the policy model establishes a training loop that enhances stability and accelerates convergence. In particular, using rollouts produced by the current policy to train both models keeps their data distributions roughly consistent, minimizing off-policy bias and covariate shift. This consistency keeps implicit step rewards calibrated to the agent's behavior, yielding dense and low-variance credit signals and preventing over- or underfitting the implicit PRM to ensure stable training. The result is a self-reinforcing loop: improved policies yield better preference data, refining the implicit PRM, which in turn delivers more accurate implicit step rewards to guide the policy.

## 3.2 Theoretical analysis

We now justify that the learning objective of our implicit PRM in Eq. 2 is a multi-turn version of standard DPO, which is equivalent to a BT model with a step-wise reward function. Formally, for any trajectory pair $\{\tau_i = \{(o_t^i, a_t^i)\}_{t=1}^{T_i}\}_{i=1}^2$ satisfying $o_1^1 = o_1^2$ (the same task prompt $x$), we have

$$\mathbb{P}(\tau_1 \succ \tau_2) = \sigma\left(\sum_{t=1}^{T_1} \beta \log \frac{\pi_\phi^*(a_t^1 \mid o_{1:t}^1, x)}{\pi_{\theta_{\text{old}}}(a_t^1 \mid o_{1:t}^1, x)} - \sum_{t=1}^{T_2} \beta \log \frac{\pi_\phi^*(a_t^2 \mid o_{1:t}^2, x)}{\pi_{\theta_{\text{old}}}(a_t^2 \mid o_{1:t}^2, x)}\right)$$

$$= \sigma\left(\sum_{t=1}^{T_1} r_\phi^*(o_{1:t}^1, a_t^1) - \sum_{t=1}^{T_2} r_\phi^*(o_{1:t}^2, a_t^2)\right) \tag{7}$$

where $\pi_\phi^*$ denotes the optimal implicit PRM. This is a multi-turn version of standard DPO similar to Rafailov et al. (2024a), but we consider each action sequence as an optimized unit in the MDP instead of the token-level MDP. In addition, we only calculate the loss on the action tokens while excluding the log probability ratios of the tokens generated by the environment. Based on this multi-turn version, Eq. 7 shows that the learning objective of our implict PRM in Eq. 2 is equivalent to a BT model with a step-wise reward function:

$$r_\phi^*(o_{1:t}, a_t) = \beta \log \frac{\pi_\phi^*(a_t \mid o_{1:t}, x)}{\pi_{\theta_{\text{old}}}(a_t \mid o_{1:t}, x)}. \tag{8}$$

This is also a variant of a token-level dpo objective which has been proved to be equivalent to a BT model with a token-wise reward function (see Zhong et al. (2025) for proof details).

## 4 EXPERIMENTS

We evaluate **iStar** across a variety of agentic tasks to demonstrate: (1) its effectiveness in training LLM agents for long-horizon reasoning and acting; (2) high sample efficiency and training stability given implicit step rewards; (3) improved exploration efficiency evidenced by increased rewards and fewer steps; and (4) the core components of **iStar** for credit assignment in agentic RL.

**Benchmarks.** We evaluate LLM agents in three challenging environments: (1) WebShop (Yao et al., 2022), a text-based web environment where the agent interacts with a HTML-based website to search, nevigate, and purchase an item given an user instruction, requiring multi-step decision making; (2) VisualSokoban (Schrader, 2018) with $6 \times 6$ size, a puzzle game where the agent has to push all boxes on targets, requiring spatial reasoning and long-term planning over both visual and textual inputs; (3) SOTOPIA (Zhou et al., 2024), an open-ended social interaction environment where the agent interacts with another LLM agent given a social scenario, role profiles and private goals, requiring reasoning over the other agent's real-time strategies. During training, we use scenarios from SOTOPIA-$\pi$ (Wang et al., 2024b). See Appendix B for more details of the environments.

**Baselines.** We compare our method against a range of competitive baselines: (1) prompting LLMs specialized in general-purpose reasoning: GPT-5 (OpenAI, 2025b), Gemini-2.5-Pro (Google, 2024), DeepSeek-R1 (Guo et al., 2025), and Claude-Sonnet-4-Thinking (Anthropic, 2025); (2) vanilla RL methods that only use outcome rewards: PPO (Schulman et al., 2017), GRPO (Shao et al., 2024), RLOO (Ahmadian et al., 2024) and REINFORCE++ (Hu et al., 2025); (3) a recent single-turn RL method: PRIME (Cui et al., 2025) that introduces token-level process rewards for policy learning; and (4) a recent agentic RL algorithm: GiGPO (Feng et al., 2025) that computes step-level advantages via same-state grouping.

**Evaluation.** For WebShop and VisualSokoban, we adopt Success Rate and Score (only for Web-Shop) as the evaluation metrics following Feng et al. (2025). These metrics are computed over validation instances and select the best score for comparison. For SOTOPIA, we report goal completion score ranging from 0 to 10, which is evaluated by GPT-4o as a proxy for human judgement following Zhou et al. (2024). We set the tempature to 0 for the LLM judge. Refer to Appendix C.2 for evaluation prompts used in each environment.

**Implementation details.** In our experiments, the implicit PRM is initialized from the base policy model by default. However, in VisualSokoban, we use Qwen2.5-VL-7B-Instruct as the base model for the policy model and Qwen2.5-7B-Instruct as the base model for the implicit PRM. We use a constant learning rate $5 \times 10^{-7}$ for the policy model and $10^{-6}$ for the implicit PRM with AdamW optimizer. Both the policy model and the implicit PRM use a batch size of 64 and micro-batch size 8. We set the advantage coefficient $\alpha = 1.0$ following (Feng et al., 2025) and $\beta = 0.05$ for the implicit PRM following (Yuan et al., 2025). Performance with varying $\alpha$ and $\beta$ is shown in Table 4. The rollout size is set to 8 per prompt. All methods share the identical RL configurations for fair comparison, and all experiments are run on $8\times$A100 GPUs. Additional training details for each environment are shown in C.1.

Table 1: **Performance on WebShop and VisualSokoban**. Qwen2.5-7B-Instruct and Qwen2.5-VL-7B-Instruct serve as the base models for the policy model in WebShop and VisualSokoban, respectively. Note that Deepseek-R1 and PPO training do not currently support multi-modal scenarios, and PRIME is only applicable to tasks with binary outcome rewards. Results are averaged over three random seeds.

| Method | WebShop | | VisualSokoban |
| --- | --- | --- | --- |
| | Success | Score | Success |
| *Prompting frontier LLMs (ReAct)* | | | |
| GPT-5 | 37.5 | 66.1 | 16.6 |
| Gemini-2.5-Pro | 30.5 | 38.4 | 16.0 |
| DeepSeek-R1 | 29.3 | 39.8 | - |
| Claude-Sonnet-4-Thinking | 35.2 | 62.0 | 19.1 |
| Base Model (ReAct) | 21.5 | 47.3 | 14.1 |
| + PPO | $78.2 \pm 4.5$ | $86.6 \pm 1.1$ | - |
| + GRPO | $80.1 \pm 1.7$ | $89.3 \pm 2.8$ | $85.6 \pm 2.8$ |
| + RLOO | $77.4 \pm 1.1$ | $87.6 \pm 4.7$ | $86.3 \pm 0.6$ |
| + REINFORCE++ | $77.0 \pm 3.9$ | $85.8 \pm 0.1$ | $81.4 \pm 8.8$ |
| + PRIME (Cui et al., 2025) | $81.5 \pm 1.8$ | $91.3 \pm 0.6$ | - |
| + GiGPO (Feng et al., 2025) | $84.1 \pm 3.9$ | $91.2 \pm 1.5$ | $85.9 \pm 2.6$ |
| **+ RLOO w/ iStar** | $\mathbf{86.5} \pm 2.8$ | $\mathbf{93.6} \pm 1.0$ | $\mathbf{91.7} \pm 1.2$ |

## 4.1 MAIN RESULTS

**Performance on benchmarks.** Table 1 showcases **iStar**'s superior performance over baselines in WebShop and VisualSokoban, with notable gains on the latter, where RL algorithms struggle with irreversible mistakes and limited foresight. Similar trends occur with much smaller base models as demonstrated in Table 5. Specifically, our method achieves state-of-the-art performance and surpasses recent multi-turn RL baseline GiGPO (Feng et al., 2025) by enabling finer-grained credit assignment, distinguishing good and bad actions with implicit rewards for each step than relying on same-state grouping. Our method also outperforms recent single-turn RL baseline PRIME (Cui et al., 2025) that uses token-level process rewards, which provides overly fine-grained rewards that complicate policy training in multi-turn RL with high-variance (see Figure 4(a)-(b)). In SOTOPIA, where GiGPO and PRIME are inapplicable due to open-ended state space and unverifiable rewards, Table 2 shows that **iStar** still achieves state-of-the-art performance. Particularly, compared to vanilla RL baselines, our method improves goal completion in hard social scenarios by 14% ($7.92 \rightarrow 8.06$) in self-chat, and up to 48% ($6.68 \rightarrow 7.16$) increase when chatting with GPT-4o. This demonstrates the generalizability of our method to a wide variety of interactive environments.

**iStar using different vanilla RL algorithms.** Since our method is compatible with various RL methods, we also evaluate it using different RL algorithms, including RLOO (Ahmadian et al., 2024), REINFORCE++ (Hu et al., 2025) and GRPO (Shao et al., 2024), and compare each to its vanilla version that only uses outcome rewards. As shown in Figure 3, **iStar** consistently improves vanilla RL methods by integrating implicit step rewards into multi-turn RL to improve credit assignment in long-horizons. For example, **iStar** with RLOO obtains substantial gains of 6.3% in success rate on both WebShop and VisualSokoban. Similar trends occur with REINFORCE++ and GRPO, demonstrating the robustness of our method to diverse RL algorithms and environments.

## 4.2 IN-DEPTH ANALYSIS

**Sample efficiency and training stability.** Figure 4 illustrates that compared to baselines, **iStar** achieves faster improvement and higher final performance in validation metrics during multi-turn RL, demonstrating superior sample efficiency. Particularly, in Figure 4(a), our method achieves the score of vanilla RLOO in WebShop in just 105 steps, around 2× improvement in training efficiency. By 165 steps, we reach the highest score of 94.7%, showcasing significant improvements in both efficiency and performance. Notably, while PRIME exhibits comparable early-stage performance

Table 2: **Performance on Sotopia.** 'Self-Chat' refers to dialogues where the model under evaluation interacts with itself, while "GPT-4o-as-Partner" denotes interactions between the model and GPT-4o. "Goal" refers to the goal completion rate (on a scale of 0-10). The "Hard" subset comprises test scenarios in SOTOPIA that require advanced reasoning capabilities, and "All" represents the complete test set. Results are averaged over three random seeds.

| Method | Self-Chat | | GPT-4o-as-Partner | |
|---|---|---|---|---|
| | Goal (Hard) | Goal (All) | Goal (Hard) | Goal (All) |
| *Prompting frontier LLMs (ReAct)* | | | | |
| GPT-5 | 7.21 | 8.95 | **7.70** | **8.90** |
| Gemini-2.5-Pro | 6.74 | 8.27 | 7.43 | 8.41 |
| DeepSeek-R1 | 6.98 | 8.56 | 7.30 | 8.44 |
| Claude-Sonnet-4-Thinking | 6.39 | 8.64 | 7.02 | 8.62 |
| Qwen2.5-7B-Instruct (ReAct) | 5.56 | 6.77 | 5.51 | 7.30 |
| + PPO | $6.63 \pm 0.24$ | $8.25 \pm 0.09$ | $6.27 \pm 0.14$ | $8.07 \pm 0.08$ |
| + GRPO | $6.97 \pm 0.24$ | $8.31 \pm 0.06$ | $6.42 \pm 0.31$ | $7.84 \pm 0.06$ |
| + RLOO | $5.70 \pm 0.16$ | $7.13 \pm 0.02$ | $6.09 \pm 0.13$ | $7.77 \pm 0.03$ |
| + REINFORCE++ | $6.17 \pm 0.30$ | $7.87 \pm 0.09$ | $6.38 \pm 0.05$ | $7.93 \pm 0.09$ |
| **+ GRPO w/ iStar** | $\mathbf{7.11} \pm 0.19$ | $\mathbf{8.42} \pm 0.03$ | $\mathbf{6.76} \pm 0.18$ | $\mathbf{8.36} \pm 0.03$ |
| Llama3.1-8B-Instruct (ReAct) | 5.89 | 6.95 | 5.82 | 7.43 |
| + PPO | $7.76 \pm 0.14$ | $9.05 \pm 0.03$ | $6.64 \pm 0.03$ | $8.14 \pm 0.01$ |
| + GRPO | $7.92 \pm 0.08$ | $9.12 \pm 0.02$ | $6.68 \pm 0.03$ | $8.14 \pm 0.02$ |
| + RLOO | $6.48 \pm 0.15$ | $8.33 \pm 0.03$ | $6.51 \pm 0.14$ | $8.02 \pm 0.06$ |
| + REINFORCE++ | $7.84 \pm 0.14$ | $9.06 \pm 0.04$ | $6.38 \pm 0.23$ | $7.99 \pm 0.10$ |
| **+ GRPO w/ iStar** | $\mathbf{8.06} \pm 0.11$ | $\mathbf{9.20} \pm 0.03$ | $\mathbf{7.16} \pm 0.14$ | $\mathbf{8.45} \pm 0.03$ |

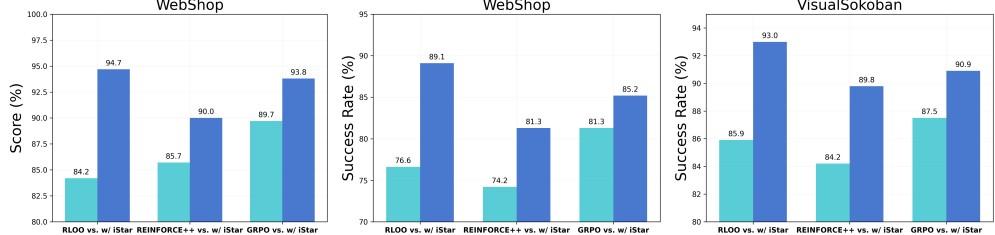

Figure 3: **Compatibility verification of iStar with various RL algorithms.** Qwen2.5-7B-Instruct and Qwen2.5-VL-7B-Instruct serve as the base models for the policy model in WebShop and VisualSokoban, respectively. Results are reported based on one random seed.

on WebShop, its growth stagnates and experiences sharp fluctuations. This is because overly fine-grained process rewards in token-level will complicate policy learning in long-horizon interactions, which usually involve much longer sequences than single-turn tasks. In contrast, **iStar** continues to improve consistently, suggesting that our implicit step rewards provide dense feedback for exploration while being sufficiently coarse to reduce variance for stable training.

With increased compute, our method's advantages in sample efficiency and training stability not only persist but become more pronounced. As shown in Figure 10 and Figure 11, its performance improvements remain consistently stable as training progresses. In contrast, baselines such as vanilla RLOO and GiGPO become increasingly unstable and can even degrade with more training steps. In addition, although our method introduces an implicit PRM that adds some computation per training step, this overhead is relatively small since policy rollouts dominate the total training cost. Our method can also reduce per-step computation owing to its shorter trajectories (see Figure 5(c)-(d)). Consequently, our method generally achieves better performance than the baselines for the same amount of GPU hours as illustrated in Figure 12.

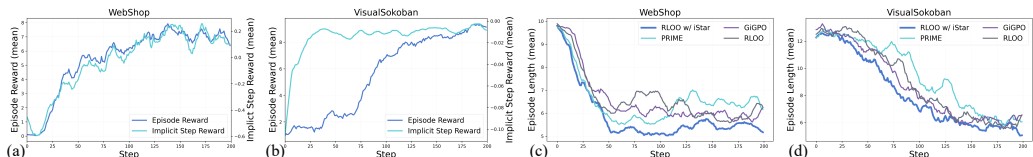

(a) (b) (c)

Figure 4: **Validation performance (10-step moving) during RL in WebShop and Visual-Sokoban.** Note that PRIME can only be applied to tasks with binary outcome rewards.

(a) (b) (c) (d)

Figure 5: **Training dynamics (10-step moving) of iStar in WebShop and VisualSokoban**. (a)-(b): Dynamics of the episode and implicit step rewards during RL training by our method. (c)-(d): The episode length versus training step compared to baselines.

**Exploration efficiency.** To demonstrate the implicit step rewards provide useful guidance for policy learning, we visualize the evolution of step- and episode-level rewards in Figure 5(a) and (b). In particular, the implicit step reward improves very early (especially in VisualSokoban) and then the episode reward follows, indicating that our method first captures good local action heuristics and then composes them into higher-return trajectories. Even when we use a smaller base model that may generate lower-quality rollouts in the early stage of RL, our implicit PRM still remains stable and effective as shown in Figure 9. A key consequence is that unnecessary actions are reduced during multi-turn interactions, resulting in shorter episode lengths. As shown in Figure 5(c) and (d), episode lengths decrease without compromising task success, as evidenced by the consistent increase in episode rewards.

Table 3: **Ablation studies on core components of iStar**. "RLOO": only outcome rewards are used to compute advantages for policy updates. "w/ environmental process rewards": use raw step rewards provided by VisualSokoban to calculate step-level advantages. "w/ merged rewards": implicit step rewards are added directly to outcome rewards before advantage computation. "w/ token-level process rewards": the implicit PRM produces rewards for each token along the entire trajectory rather than step-level rewards for each action sequence. Results are reported using one seed.

| Method | WebShop | | VisualSokoban |
|---|---|---|---|
| | Success | Score | Success |
| RLOO | 76.6 | 84.2 | 85.9 |
| w/ environmetal process rewards | - | - | 87.5 |
| w/ merged rewards | 81.3 | 90.7 | 88.3 |
| w/ token-level process rewards | 82.0 | 90.0 | 89.1 |
| **w/ iStar** | **89.1** | **94.7** | **93.0** |

## 4.3 ABLATION STUDIES

Table 3 presents results of ablation experiments to validate the necessity of key components of our method for effective credit assignment. First, raw step penalties provided in VisualSokoban show limited improvement over vanilla RL, suggesting that the implicit step rewards learned by **iStar** are superior credit signals for policy learning. Second, merging the implicit step rewards into episode rewards obtains gains over vanilla RL but the improvement are modest compared to our method.

This indicates that we should not only reward intermediate actions but also gate credit by final task success to prevent speculative reward exploitation. Therefore, combining signals at the advantage level is crucial to credit assignment in long-horizons. Third, learning token-level process rewards is sub-optimal to multi-turn RL, suggesting that overly fine-grained rewards may introduce noise and thus increase the difficulty of policy learning.

We also compare our method to training the policy on SOTOPIA using multi-turn DPO alone. As shown in Table 6, vanilla GRPO already substantially outperforms both online and offline multi-turn DPO. Building on this, our method further improves performance by using both step and episode rewards, enabling more effective credit assignment in multi-turn RL.

## 5 RELATED WORK

PRMs has been widely explored in single-turn tasks, such as mathematical reasoning and single-shot code generation. In these settings, PRMs are used to score intermediate steps for test-time search or reranking (Lightman et al., 2024; Wang et al., 2024a; Mahan et al., 2024), or for online-RL (Dou et al., 2024; Setlur et al., 2025; Zha et al., 2025). Moving to dynamic, interactive tasks, process rewards are usually constructed in three ways: (1) handcrafted step rewards assigned to tool execution (Zeng et al., 2025) or meta-reasoning tags (Zhang et al., 2025b); (2) GRMs (Liu et al., 2025; Zou et al., 2025) that label step quality; (3) and implicit PRMs (Yuan et al., 2025) that produce token-level process rewards.

However, manually-designed or judge-based step labels are costly and biased, suffering from reward hacking. Learning step Q-values (Choudhury, 2025) can reduce the bias but a fixed PRM may poorly estimate Q-values for unseen actions during inference. PRIME(Cui et al., 2025) partially addresses these issues by jointly training an implicit PRM with the generator. Yet, with overly fine-grained rewards in token-level, it introduces noises and destabilizes training in multi-turn RL. Another issue with PRIME is that it applies a cross-entropy loss to optimize the implicit PRM, which is only applicable to tasks with binary outcome rewards. Instead of learning a PRM, Feng et al. (2025) addresses credit assignment by computing step-level advantages via same-state grouping. While effective in tasks with finite state-action space, it relies on exact state overlaps and cannot be generalized to open-ended language environments where same state is rare. In contrast, our method learns implicit rewards at the step-level through a multi-turn DPO objective, offering label-efficient step rewards with low-variance in long-horizon RL and showing robustness and generalization to (un)verifiable rewards even in open-ended environments.

## 6 CONCLUSION AND DISCUSSION

We propose **iStar**, a general credit-assignment strategy for LLM agents. In particular, we alternatively optimize an implicit PRM with the policy model, which provides implicit step rewards to guide policy learning and results in a self-reinforcing training loop. To optimize the implicit PRM in an online setting, we propose a multi-turn DPO objective that is guaranteed to be equivalent to a BT model with a step-wise reward function. In practice, our method can be plugged into a variety of RL algorithms and generalized to (un)verifiable rewards even in open-ended environments. Empirical results show that **iStar** achieves averaged 86.5% success and 93.6% score on WebShop, and reaches 91.7% success on VisualSokoban. In SOTOPIA, our method improves goal completion in hard scenarios by up to 14% in self-chat and 48% against GPT-4o.

In the future, our method can also be validated in math problems or code generation to provide implicit step rewards for intermediate CoT steps. It can also be applied to test-time scaling for search guidance. Regarding the limitations, we currently separate the implcit PRM from the policy model during training, which however, can be a unified model trained with different objectives to reduce computation memory and potentially improve representation sharing. Additionally, in SOTOPIA, our implicit PRM is trained to only predict goal-completion preferences, while future work could be extended to multi-objective implicit PRMs.

ETHICS STATEMENT

The development of autonomous LLM agents capable of independent thinking and decision-making in interactive environments, including those involving human users, offer transformative potential to real-world applications but could also introduce some ethical concerns. First, if improperly designed or monitored, such an agent may exhibit behaviors that prioritize success metrics at the expense of human values and safety. For instance, an LLM agent operating in dynamic interactions may inadvertently take actions that conflict with ethical principles, reinforce biases, or neglect the broader societal and interpersonal impacts of its decisions. These risks highlight the need for integrating rigorous safeguards to ensure alignment with human-centric values. Second, a commitment to transparency, accountability, and inclusivity in the training and evaluation of these LLM agents is essential to ensuring they exhibit empathy, integrity, and fairness in reasoning and acting. By embedding ethical principles into the development process, it is expected to mitigate potential harms and harness the potential of these technologies for the benefit of individuals and the society.

REPRODUCIBILITY STATEMENT

To reproduce our experiments, we provide the source code at https://github.com/Tongyi-ConvAI/Qwen-Character/tree/main/CharacterRL-iStar. Regarding the theoretical analysis in Section 3.2, since recent work (Zhong et al., 2025) has provided a detailed proof in a token-wise setup, which is a variant of our step-wise justification, we do not elaborate more in this paper. For specific proof details, please refer to Zhong et al. (2025). With respect to training data, our training pipeline relies solely on initial task prompts, as the policy model autonomously generates online rollouts during interactions with the environment (including a user simulator in SOTOPIA). Therefore, we do not require a fixed offline dataset or a dedicated data processing step. See Appendix C.2 for the detailed task prompt templates used in our experiments.

ACKNOWLEDGEMENTS

This work is supported by National Key R&D Program of China (2025ZD0122003), National Key Laboratory under Grant 241-HF-D08, and in part by the Beijing Major Science and Technology Project under Contract no. Z251100008425009.

We gratefully acknowledge the Qwen-Character Team at Tongyi Lab for their support in providing computing resources and funding for this work.

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

# A  ALGORITHM

The algorithm flow of iStar is detailed in Algorithm 1.

---

**Algorithm 1:** Training LLM Agents with iStar (GRPO as an example)

---

**Input:** Task distribution $p(X)$, language model $\pi_{\theta_{\text{init}}}$, outcome reward verifier or model $r_o$,
      training steps M, rollout size N, mixing weight $\alpha$
**Output:** Optimized policy $\pi_\theta$ and PRM $\pi_\phi$
Initialize policy model $\pi_\theta \leftarrow \pi_{\theta_{\text{init}}}$, $\pi_{\theta_{old}} \leftarrow \pi_{\theta_{\text{init}}}$, PRM $\pi_\phi \leftarrow \pi_{\theta_{\text{init}}}$;
**for** iteration $= 1, ..., M$ **do**
    // Multi-step rollouts collection
    Sample task $x \sim p(X)$ and initialize N identical environments
    **for** $t = 1, ..., T$ **do**
        Sample actions $\{a_t^i \sim \pi_\theta(o_{1:t}^i, x)\}_{i=1}^N$
        Execute actions and observe next observation $\{o_{t+1}^i\}_{i=1}^N$
    // PRM training
    Compute outcome rewards for N trajectories: $r_o(\tau_{1:N})$
    Forward pass $\pi_\phi$ based on trajectory preferences to obtain step reward $r_\phi(a_t^i)$ with Eq. 1
    Update PRM $\pi_\phi$ on trajectories using a DPO-style objective in Eq. 2
    // Policy training
    Compute episode-level advantages $A^E(\tau_i)$ using $r_o(\tau_i)$ via Eq. 3
    Compute step-level advantages $A^S(a_t^i)$ using $r_\phi(a_t^i)$ via Eq. 4
    Combine advantages: $A(a_t^i) = A^E(\tau_i) + \alpha A^S(a_t^i)$
    Update policy $\pi_\theta$ by maximizing objective in Eq. 6
    Update old parameters: $\theta_{old} \leftarrow \theta$

---

# B  ENVIRONMENT DETAILS

**WebShop.** WebShop (Yao et al., 2022) simulates online shopping tasks on an e-commerce platform, where the agent's objective is to interpret human-provided text instructions and purchase a product that aligns with the given specifications. To accomplish this, the agent must interact with the website's search engine, select items to review from the search results, examine their descriptions and details, and choose relevant options (e.g., size, color) before finalizing the purchase by clicking the "Buy" button. To identify the best product that fulfills the user's requirements, the agent may need to compare multiple products, navigate back and forth between pages, and conduct additional searches if necessary. The environment includes over one million products sourced from amazon.com, more than 12,000 crowd-sourced instructions, and a rich set of semantic actions. At the end of each interaction, a binary reward is provided to indicate whether the task is successfully completed. The final score for each task is automatically calculated using programmatic matching functions that evaluate attributes, type, options, and price of the selected product.

**VisualSokoban.** Sokoban (Schrader, 2018) consists of rooms composed of five key elements: walls, floors, boxes, box targets, and an agent. These elements may exist in different states depending on whether they overlap with a box target. Rooms are randomly generated, which helps prevent models from overfitting to specific predefined layouts. The game includes two primary actions, Push and Move, which can be performed in four directions: Up, Down, Left, and Right. The Move action allows the agent to proceed to an empty space in the specified direction, provided there is no wall or box blocking the path. The Push action attempts to move an adjacent box, but only if the field behind the box is empty; chain pushing of multiple boxes is not allowed. If no box is adjacent, the Push action functions identically to the Move action in the same direction. Successfully completing the game by pushing all boxes onto their targets yields a reward of 10 points on the final step. Additionally, pushing a box onto a target grants a reward of 1 point, while removing a box from a target results in a penalty of -1 point. Each step incurs a small penalty of -0.1 points to discourage trajectories with many steps. VisualSokoban renders visuals in RGB, with the pixel size equal to the grid size.

**SOTOPIA.** SOTOPIA (Zhou et al., 2024) is a general-domain, open-ended platform to simulate social interactions between LLM agents. The scenarios span a diverse array of social interaction types, such as negotiation, exchange, collaboration, competition, accommodation, and persuasion. A particularly challenging subset, known as SOTOPIA-hard, involves scenarios requiring advanced strategic reasoning. Each agent is defined by character profiles, encompassing attributes like name, gender, personality, and profession. At the end of each dialogue, agents are evaluated by GPT-4o as a proxy for human judgment across seven dimensions: Goal Completion, Believability, Knowledge, Secret, Relationship, Social Rules, Financial and Material Benefits. SOTOPIA-$\pi$ (Wang et al., 2024b) is a follow-up work that uses GPT-4 to generate a new set of scenarios. The social tasks in SOTOPIA-$\pi$ are guaranteed to be entirely distinct from those in SOTOPIA.

## C  EXPERIMENT DETAILS

### C.1  TRAINING DETAILS

**WebShop and VisualSokoban.** We use Qwen2.5-(VL)-7B-Instruct (Yang et al., 2024) as the base models for policy learning. To address invalid actions produced by LLM agents, a reward penalty of -0.1 is imposed. The maximum response length is 512 tokens, while the maximum prompt length is 4096 tokens in WebShop and 1024 tokens in VisualSokoban. We sample 16 different groups per rollout in WebShop, resulting in a total of $16 \times 8 = 128$ environments. In VisualSokoban, we sample 32 different groups per rollout, resulting in a total of $32 \times 8 = 256$ environments. Instead, PPO uses 128 and 256 separate environments for rollouts in WebShop and VisualSokoban, respectively. The rollout temperature is set to 1.0, while the validation temperature is set to 0.4. We implement experiments in veRL (Sheng et al., 2024), each for 200 training steps.

**SOTOPIA.** We use Qwen2.5-7B-Instruct (Yang et al., 2024) and Llama3.1-8B-Instruct (Meta, 2024) as the base models for policy learning to demonstrate the robustness of our method to different model backbones. The maximum prompt length is 6144 tokens and the maximum response length is 2048 tokens. As with WebShop, we sample 16 different groups per rollout in WebShop, resulting in a total of $16 \times 8 = 128$ environments (PPO uses 128 separate environments). The rollout temparature is set to 0.7. Each experiment implemented in veRL consists of 800 training steps.

| $\alpha$ | $\beta$ | Success | Score |
|---|---|---|---|
| 0.5 | 0.05 | 79.7 | 89.4 |
| 0.8 | 0.05 | 84.4 | 91.3 |
| 1.0 | 0.05 | **89.1** | **94.7** |
| 1.2 | 0.05 | 85.9 | 92.6 |
| 2.0 | 0.05 | 78.1 | 90.7 |
| 1.0 | 0.1 | 82.8 | 91.8 |
| 1.0 | 0.5 | 85.2 | 94.7 |

Table 4: **Performance of our method on WebShop with varying $\alpha$ and $\beta$.** Qwen2.5-7B-Instruct is used as the base model.

### C.2  EVALUATION PROMPTS

We use ReAct (Yao et al., 2023) as the prompting strategy, with chain-of-thought (Wei et al., 2022) generated before each action. The prompt templates used for evaluating LLM or multimodal large language model (MLLM) agents in the environments are presented in Figure 6-8. Placeholders enclosed in curly braces({}) represent semantic slots, which are dynamically populated at runtime.

---
**Prompt template for WebShop**

You are an expert autonomous agent operating in the WebShop e-commerce environment.
Your task is to: {task_description}.
Prior to this step, you have already taken {step_count} step(s).
Below are the most recent {history_length} observations and the corresponding actions you took: {action_history}
You are now at step {current_step} and your current observation is: {current_observation}.
Your admissible actions of the current situation are:
[
{available_actions}
].
Now it's your turn to take one action for the current step.
You should first reason step-by-step about the current situation,
then think carefully which admissible action best advances the shopping goal.
This reasoning process MUST be enclosed within <think> </think> tags.
Once you've finished your reasoning, you should choose an admissible action for current step,
and present it within <action> </action> tags.

---

Figure 6: Evaluation prompts used for the LLM agent in WebShop.

---
**Prompt template for VisualSokoban**

You are an expert agent operating in the Sokoban environment.
Your goal is to push all the boxes onto the target spots. Once all boxes are on the targets, you win!

# Rules
You can only push boxes. You can't pull them, so plan ahead to avoid getting stuck.
You can't walk through or push boxes into walls.
To avoid traps, do not push boxes into corners or against walls where they can't be moved again.

# Visual Elements in the Image:
Character: A small, green alien-like figure with two antennae and black eyes. It represents you.
Box: A yellow crate marked with an orange "X" across its front. It is the box you need to push.
Target: A black tile outlined in red, with a small red diamond shape in the center.
It marks the destination where a box should be pushed.

# Current Step
Your current observation is shown in the image: <image>
Your admissible actions are ["up", "down", "left", "right"].

Now it's your turn to make a move (choose ONE action only for the current step).
You should first reason step-by-step about the current situation — observe the positions of boxes and targets,
plan a path to push a box toward a target, and avoid traps like corners or walls.
This reasoning process MUST be enclosed within <think> </think> tags.
Once you've finished your reasoning, you should choose an admissible action for current step,
and present it within <action> </action> tags.

---

Figure 7: Evaluation prompts used for the MLLM agent in VisualSokoban.

# D ADDITIONAL RESULTS

## D.1 RESULTS ON SMALLER LLMS

Table 5 presents results of our method compared to baselines on WebShop using Qwen2.5-1.5B-Instruct (Yang et al., 2024) as the base model. The results show that our method can also benefit base models with weaker reasoning capabilities. Furthermore, Figure 9 demonstrates that even the weaker base model generates low-quality rollouts (indicated by low episode rewards) in the early stage of RL, our implicit PRM still remains stable and effective with steadily increasing step rewards and episode rewards throughout the training.

---

**Prompt template for SOTOPIA**

Imagine you are {agent}, your task is to act/speak as {agent} would, keeping in mind {agent}'s social goal.
You can find {agent}'s goal (or background) in the 'Here is the context of this interaction' field.
Note that {agent}'s goal is only visible to you.
You should try your best to achieve {agent}'s goal in a way that align with their character traits.
Additionally, maintaining the conversation's naturalness and realism is essential (e.g., do not repeat what other people has already said before).

You are at Turn {turn}. You can say something to interact or just say 'left the conversation' to stop continuing.
Note: You can 'left the conversation' if 1. you have achieved your social goals, 2. this conversation makes you uncomfortable, 3. you find it uninteresting or you lose your patience, 4. for other reasons you want to leave.

You should first reason step-by-step to reflect on the current state of the dialogue, then think carefully what communication and social strategies best advances your goal.
This reasoning process MUST be enclosed within <think> </think> tags.
Once you've finished your reasoning, provide your response and present it within <action> </action> tags.

---

Figure 8: Evaluation prompts used for LLM agents in SOTOPIA.

Table 5: **Performance on WebShop with Qwen2.5-1.5B-Instruct as the base model.** Results are reported using one seed.

| Method | Success | Score |
|---|---|---|
| Qwen2.5-1.5B-Instruct (ReAct) | 10.4 | 46.1 |
| + RLOO | 71.9 | 85.7 |
| + GiGPO | 72.7 | 86.8 |
| + PRIME | 74.2 | 86.6 |
| **RLOO w/ iStar** | **80.5** | **91.5** |

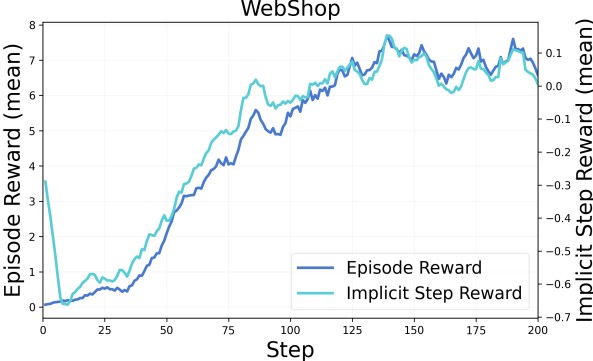

Figure 9: **Training dynamics (10-step moving) of iStar using Qwen2.5-1.5B-Instruct as the base model on WebShop**. The left vertical axis illustrates the trend of trajectory-level outcome rewards during RL, while the right vertical axis represents the progression of implicit step rewards.

## D.2 RESULTS WITH MORE COMPUTE

Figures 10 and 11 show that the sample-efficiency and stability advantages of our method not only persist but grow with increased compute. As training steps increase, our method yields larger and more consistent performance gains, whereas baselines such as vanilla RLOO and GiGPO become increasingly unstable and can even degrade in performance over time.

Regarding GPU hours (Figure 12), our method still outperforms the baselines in both sample efficiency and training stability, even though the implicit PRM adds some computation per training

step. This overhead is relatively small because policy rollouts dominate the total training cost, and our shorter trajectories can further reduce per-step computation (see Figure 5(c)–(d)).

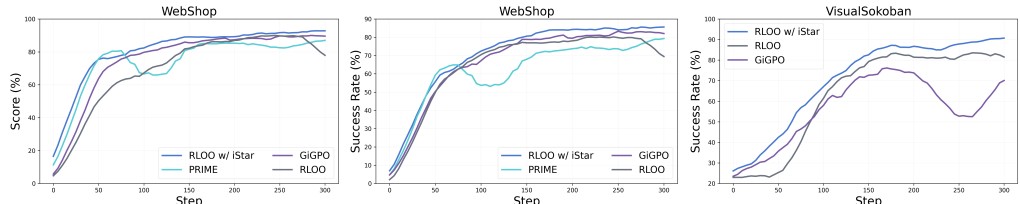

Figure 10: Validation performance comparison (10-step moving) on WebShop and VisualSokoban with more training steps.

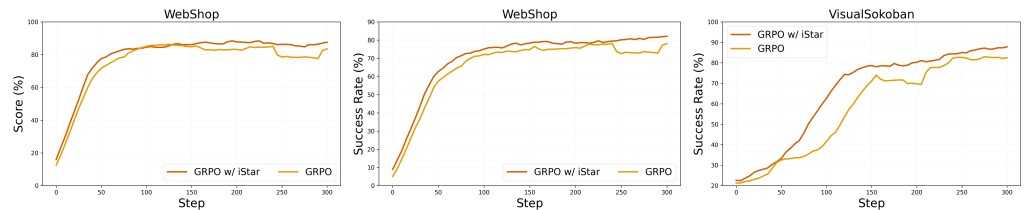

Figure 11: Validation performance comparison (10-step moving) between iStar and vanilla GRPO on WebShop and VisualSokoban.

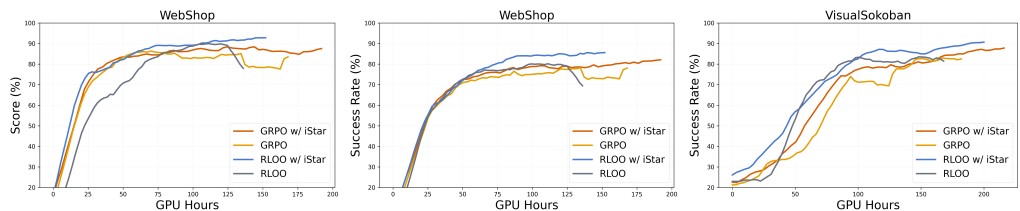

Figure 12: Validation performance comparsion (10-step moving) on WebShop and VisualSokoban with respect to GPU hours.

Table 6: **Performance comparison between iStar and DPO in self-chat settings of SOTOPIA.** Llama3.1-8B-Instruct is used as the base model. We apply supervised fine-tuning (SFT) before multi-turn DPO, whereas GRPO and iStar do not require this step.

| Method | Goal (Hard) | Goal (All) |
|---|---|---|
| SFT + DPO (offline) | 6.40 | 8.29 |
| SFT + DPO (online) | 6.68 | 8.46 |
| GRPO | 7.92 | 9.12 |
| **GRPO w/ iStar** | **8.06** | **9.20** |

### D.3 PERFORMANCE COMPARISON AGAINST USING MULTI-TURN DPO TO OPTIMIZE THE POLICY MODEL

To evaluate our method—which integrates an implicit PRM optimized via DPO into online RL—we compare it against using multi-turn DPO alone to train the policy on SOTOPIA. In the offline setting, we use the datasets proposed by Kong et al. (2025) and follow Rafailov et al. (2024a); Tajwar et al. (2025) to perform multi-turn DPO at the trajectory level. For online DPO, trajectory preference pairs are constructed online using outcome rewards produced by the LLM-as-judge in SOTOPIA, and optimized by multi-turn DPO following Guo et al. (2024).

Table 7: **Performance comparison on WebShop with and without the KL-divergence penalty.** Qwen2.5-7B-Instruct is used as the base model.

| Method | Success | Score |
|---|---|---|
| **RLOO w/ iStar** | **89.1** | **94.7** |
| RLOO w/ iStar & KL | 82.8 | 92.0 |

Results in Table 6 shows that vanilla GRPO already significantly outperforms both online and offline multi-turn DPO. Our method further improves performance by leveraging both step and episode rewards to achieve better credit assignment in multi-turn RL.

## D.4 PERFORMANCE COMPARISON WITH AND WITHOUT KL-DIVERGENCE PENALTY

Table 7 compares performance on WebShop with and without the KL-divergence penalty. Dropping the KL penalty improves performance, supporting our hypothesis that removing it allows the policy to better explore the action space in LLM agent tasks.

