# OpenReview forum: "Agentic Reinforcement Learning with Implicit Step Rewards"
_ICLR.cc/2026/Conference — ICLR 2026 Poster_

### Official Review · Reviewer_m1yj · 2025-10-21

**Soundness:** 3
**Presentation:** 3
**Contribution:** 2
**Rating:** 4
**Confidence:** 4

**Summary:**

Online Reinforcement Learning (RL) has recently emerged as a powerful technique to improve LLMs’ capabilities in a wide range of tasks ranging from single step reasoning problems like math or coding to multi-turn agentic tasks such as web navigation/tool use. For multi-turn tasks where agents need to interact with an environment multiple times to successfully complete their objective, online RL is a promising approach. A key problem in this scenario is credit assignment — many tasks often have sparse or trajectory level reward only available at the end of the trajectory, and this makes it difficult to learn whether the intermediate actions taken by the agent are good or bad. This paper proposes a method for learning implicit step level rewards and leverage it for online RL training. Specifically, the proposed method, iStar, uses DPO-style log-likelihood ratio as an implicit reward for intermediate steps, and combines it with trajectory level reward (or advantage) to produce per step learning signal. The method is validated on multi-turn tasks like WebShop and VisualSokoban, showing gains over online RL methods using just trajectory level reward signal.

**Strengths:**

1. The proposed method is interesting. There is prior work that has used the DPO-style log-likelihood ratio as reward (and advantage), but I have not seen it being used as advantage in an online RL framework.

2. The paper is well-written and easy to follow.

**Weaknesses:**

(**Results are not strong enough**)

The gains we see in the method are not significant enough (in my opinion) compared to the level of complexity introduced in the paper. Specifically, if we focus on the results presented in Figure 4, the difference between the much simpler RLOO baseline and iStar is not significant enough. Moreover, to me it seems like the RLOO validation curve still has a positive gradient near the end of training, and hence more training steps might flip the order of performance between RLOO and iStar.

Moreover, RLOO is a less commonly used RL algorithm. Could the authors present the learning curves from GRPO and GRPO with iStar?

(**Comparison at more training steps**)

Based on the webshop and visual sokoban results in Figure 4, could the authors train for 300 steps (instead of 200) for all the baselines? I am not convinced that the gains may hold as one puts more compute in the baseline vs the proposed method, and would like to see longer training curves.

(**Comparison at GPU hours/FLOP level**)

Due to the training of the implicit PRM, which I believe is a separate model (correct me if I am wrong), iStar would spend significantly more compute compared to regular GRPO/RLOO. Hence, performance comparison at gradient step level is not a fair comparison. I would require a plot with

1. X-axis: GPU hours or total FLOPs

2. Y-axis: Validation performance

Where iStart shows superior performance over the baselines in order to increase my score of this paper.

(**Why does PPO perform so poorly?**)

In Table 1, PPO performs notably poorly, despite using the same idea — a trained critic function to estimate step level advantage. Could the authors explain this result?

**Questions:**

1. The implicit PRM is parametrized by $\phi$ (denoted by $\pi_\phi$), whereas the policy is parametrized by $\theta$ (denoted by $\pi_\theta$). In practice, what do these symbols mean? What model is used as the implicit PRM? Is $\phi$ different from $\theta$?

2. Why not just perform multi-turn DPO using trajectory level preference pairs constructed using task success/outcome level reward, but in an online fashion? Prior works [1, 2] has already used some variant of multi-turn DPO for sequential decision making tasks, albeit in the offline manner. I would like to see both the online/offline version of this as a baseline, and compare against [1, 2].

# References

[1] Your Language Model is Secretly a Q-Function, https://arxiv.org/abs/2404.12358

[2] Training a Generally Curious Agent, https://arxiv.org/abs/2502.17543

---

> ### Author Response · Authors · 2025-11-22
> **Explanations on performance gains and added experimental results**
>
> We are very grateful for the detailed feedback and constructive suggestions from Reviewer m1yj. We are encouraged that Reviewer m1yj finds our method interesting and paper well-written. We have added additional experimental results and discussions as suggested. Below we address all the questions and concerns in detail.
>
> ---
>
> > **Q1:** The gains we see in the method are not significant enough (in my opinion) compared to the level of complexity introduced in the paper.
>
> **A1:** 1. We would like to emphasize that **our method achieves state-of-the-art performance on all the three benchmarks** we studied:
> - On WebShop, our method surpasses vanilla RLOO by 9.1\% ($77.4\rightarrow86.5$) in the success rate and 6\% ($87.6\rightarrow93.6$) in the score. (see Table 1)
> - On VisualSokoban, our method improves the success rate by 5.4\% ($86.3\rightarrow91.7$) over vanilla RLOO. (see Table 1)
> - On SOTOPIA, our method improves the goal completion score in hard social scenarios over vanilla GRPO by 14\% ($7.92\rightarrow8.06$) in self-chat, and by up to 48\% ($6.68\rightarrow7.16$) when chatting with GPT-4o. (see Table 2)
>
> 2. Performance gains should not be viewed from Figure 4, as it is **a smoothed result of raw data**.
> - Results in Figure 4 are smoothed using a 10-step moving average to better illustrate training progress and compare overall trends with the baselines. Consequently, the value at any specific step in Figure 4 will differ from the corresponding raw value.
> - For rigorous numerical comparisons of performance gains, please refer to Tables 1 and 2, which report the best validation performance averaged across three seeds.
>
> 3. We have added **Figure 10** in Appendix D.2 to show the learning curves of our method compared to baselines with more training steps. (see **A2**)
> 4. We have added **Figure 11** in Appendix D.2 to present the learning curves of GRPO and GRPO w/ iStar in 300 training steps. Results shows that our method achieves higher sample efficiency and training stability than vanilla GRPO, especially in VisualSokoban.
>
> > **Q2:** Based on the webshop and visual sokoban results in Figure 4, could the authors train for 300 steps (instead of 200) for all the baselines?
>
> **A2:** Thanks for the suggestion! We have **added Figure 10 and Figure 11 in Appendix D.2** to show the smoothed validation performanc (10-step moving) in 300 training steps.
> - Our gains in sample efficiency and training stability persist and even strengthen with more compute. As training steps increase, our method delivers increasingly pronounced and consistently stable performance improvements.
> - In contrast, baselines (e.g.,vanilla RLOO and GiGPO) show growing instability and even performance degradation as training continues.
> - Notably, the iStar validation curve still has a positive slope near the end of 300 training steps, whereas the RLOO curve exhibits a negative slope near the end of training.
>
> > **Q3:** Due to the training of the implicit PRM, which I believe is a separate model (correct me if I am wrong), iStar would spend significantly more compute compared to regular GRPO/RLOO. Performance comparison at GPU hours or total FLOPs is required.
>
> **A3:** Great suggestion! The implicit PRM is a separate model and does consume more computation per training step compared to vanilla GRPO/RLOO. However, the policy model’s rollouts dominate the total training cost, so the additional overhead from the PRM is relatively small.
>
> In addition, our method can reduce per-step computation time owing to its shorter trajectories (see Figure 5c-d). Below we **list the GPU hours** required for different methods on WebShop and VisualSokoban.
>
> | Method | WebShop | VisualSokoban |
> | --- | --- | --- |
> | RLOO | 136h | 168h |
> | RLOO w/ iStar | 152h | 200h|
> | GRPO | 168h | 182h |
> | GRPO w/ iStar | 192h | 216h |
>
> Results above show that our method requires only about $1.1\times$ the training time of vanilla GRPO/RLOO on WebShop and $1.2\times$ on VisualSokoban.
>
> We have also **added Figure 12 in Appendix D.2** to show the smoothed validation performance (10-step moving) on WebShop and VisualSokoban with respect to GPU hours.
> Results show that in general, our method achieves superior performance over baselines in sample efficiency and training stability.

---

> ### Author Response · Authors · 2025-11-22
> **Explantions about PPO performance and implicit PRMs**
>
> > **Q4:** In Table 1, PPO performs notably poorly, despite using the same idea — a trained critic function to estimate step level advantage. Could the authors explain this result?
>
> **A4:** We would like to first clarify that our method does not involve a critic model, and the implicit PRM in our method functions differently from the critic in PPO. Then we explain why PPO performs poorly in our task setting.
>
> 1. The implicit PRM in our method and the critic model in PPO **differ in their usage for advantage estimation and their training objectives**.
> - Our implicit PRM predicts step rewards for each action, while the critic model in PPO predicts the state value for each token.
> - The step rewards predicted by our implicit PRM is used to compute step-level advantages via group relative normalization (e.g., GRPO/RLOO/Reinforce++). Instead, PPO uses the token-level values to compute advantages via GAE.
> - Our implicit PRM is optimized via multi-turn DPO based on trajectory preferences, while PPO fits its value function (learned by the critic) to discounted returns based on Bellman backup:$V_\pi \leftarrow r(o_t,a_t)+\gamma V_\pi(o_{t+1})$.
>
> 2. The main reason why PPO performs poorly is that **the critic model is trained only on outcome rewards**.
> - In our task setting, PPO's critic has to learn token-level values from very sparse and delayed rewards in long trajectories. This means only the reward for the last token in the trajectory is assigned the outcome reward; the rewards for all other tokens are 0. This makes its advantages high‑variance and biased, leading to less stable and sample‑inefficient learning.
> - Instead, our method sidesteps value‑function approximation and learns an implciit PRM to provide dense, low‑variance step rewards that PPO’s critic does not have access to.
>
> > **Q5:** In practice, what do $\phi$ and $\theta$ mean? What model is used as the implicit PRM? Is $\phi$ different from $\theta$?
>
> **A5:** Sorry for this misunderstanding. $\phi$ denotes the parameters of the implicit PRM, and $\theta$ denotes the parameters of the policy model.
>
> In our experiments, the implicit PRM is initialized from the base policy model by default. For example, in WebShop, we use Qwen2.5-7B-Instruct as the backbone for both the implicit PRM and the policy model. However, in VisualSokoban which is a multi-modal environment, we use Qwen2.5-VL-7B-Instruct as the backbone for the policy model to handle vision-language inputs, and use Qwen2.5-7B-Instruct as the backbone for the implicit PRM. We have **moved these implementation details to the main paper** (see Section 4).
>
> Therefore, the initial $\phi$ is the same as the initial $\theta$. But as training progresses, they become different as they are optimized in different directions due to the distinct training objective of the implicit PRM (see Eq.2) and the policy model (see Eq.6).

---

> ### Author Response · Authors · 2025-11-22
> **Added results of multi-turn DPO**
>
> > **Q6:** Why not just perform multi-turn DPO using trajectory level preference pairs constructed using task success/outcome level reward, but in an online fashion? I would like to see both the online/offline version of this as a baseline, and compare against [1, 2].
>
> **A6:** 1. We do not perform multi-turn DPO to train the policy model due to **our research question for agentic RL** (see line 57-59 in Section 1).
> - Because multi-turn DPO, even when done online, assigns credit only at the trajectory level, which may lead to high‑variance, unstable learning and poor exploration in long‑horizon agent tasks.
> - Our method uses multi-turn DPO **only as a tool to learn a step‑wise implicit reward function**, then combines step‑ and episode‑level advantages in standard RL updates. This yields dense, lower‑variance credit assignment that scales to multi‑turn, open‑ended environments, which prior multi‑turn DPO‑style methods do not provide.
>
> 2. We have **conducted experiments of online and offline multi-turn DPO** on SOTOPIA in the self-chat setting and the results are shown below.
> - We first perform supervised fine-tuning (SFT) before DPO due to the extremely poor performance of DPO without SFT (however, our method does not require SFT before RL).
> - We use the datasets from [1] to perform offline multi-turn DPO, and use Llama3.1-8B-Instruct as the base model across experiments.
> - The implementation of offline multi-turn DPO is exactly the same as [2], which uses a multi-turn version of DPO introduced in [3]. However, [3] performs the multi-turn DPO at the token level (each token as an action in a single-turn response) rather than the step level (a sequence of tokens as an action in a trajectory).
> - The implementation of online multi-turn DPO follows [4].
>
> | Method | Goal (hard) | Goal (all) |
> | --- | --- | --- |
> | SFT+DPO (offline) | 6.40 | 8.29 |
> | SFT+DPO (online) | 6.68 | 8.46 |
> | GRPO | 7.92 | 9.12 |
> | GRPO w/ iStar | **8.06** | **9.20** |
>
> The results above show that vanilla GRPO has already achieved significant improvements over online and offline DPO. Using our method, the performance continues to improve as we utilize both step and episode rewards for better credit assignment in multi-turn RL.
>
> ---
>
> Thanks again for your constructive suggetions and efforts in reviewing our paper. We hope the results and explanations above have addressed all the questions and concerns. We are happy to discuss if there are any other questions.
>
> ---
>
> References:
>
> [1] Kong, Aobo, et al. "Sdpo: Segment-level direct preference optimization for social agents." ACL 2025.
>
> [2] Tajwar, Fahim, et al. "Training a generally curious agent." arXiv preprint arXiv:2502.17543 (2025).
>
> [3] Rafailov, Rafael, et al. "From $ r $ to $ q^* $: Your language model is secretly a q-function." COLM 2024.
>
> [4] Guo, Shangmin, et al. "Direct language model alignment from online ai feedback." arXiv preprint arXiv:2402.04792 (2024).

---

> > ### Comment · Reviewer_m1yj · 2025-11-22
> >
> > All my concerns have been addressed, and I have increased my score from 4 to 6!
> >
> > Kudos to the authors, and I would request them to update their papers with all new references and additional results during the rebuttal period if possible.

---

> > > ### Author Response · Authors · 2025-11-23
> > >
> > > Thanks for your response and increase of the score!
> > >
> > > We have updated our paper with all new references and additional results (the modifications in the paper are highlighted in blue).
> > >
> > > Thank you again for your time and timely feedback!

---

### Official Review · Reviewer_rhUJ · 2025-10-27

**Soundness:** 3
**Presentation:** 3
**Contribution:** 3
**Rating:** 6
**Confidence:** 3

**Summary:**

This paper proposes iStar, a general credit-assignment strategy for LLM-based agents in reinforcement learning. The method introduces implicit step rewards, which provide dense yet stable feedback signals without explicit step-level annotations. iStar jointly trains a policy model (the LLM agent) and an implicit process reward model (PRM) through trajectory-based direct preference optimization (DPO). The PRM generates implicit step rewards by comparing current actions to those from a previous policy snapshot, providing step-wise credit signals that are aggregated with episode-level advantages. The authors theoretically show that this learning objective is equivalent to a Bradley–Terry model with a step-wise reward function. Empirical results on WebShop, VisualSokoban, and SOTOPIA demonstrate that iStar improves performance and sample efficiency over several reinforcement learning baselines (PPO, GRPO, RLOO, REINFORCE++) and achieves state-of-the-art results. Ablations confirm that implicit step rewards enhance exploration and stability compared to token-level or handcrafted process rewards.

**Strengths:**

1. The integration of implicit step rewards with trajectory-level DPO represents a creative approach to balancing dense feedback with low variance in long-horizon RL. It avoids the pitfalls of both sparse end-of-trajectory rewards and noisy token-level signals.
2. The method achieves consistent improvements across diverse benchmarks, and this indicates good adaptability of the framework.
3. A solid theoretical proof is provided to ground the method. The equivalence proof connecting the DPO objective to a step-wise Bradley–Terry model adds credibility to the approach and helps ground the intuition behind.

**Weaknesses:**

1.  The framework’s performance likely depends on the capability of the base model, the nature of the task environment, and the quality of outcome reward verifiers. These dependencies are not theoretically analyzed, and the paper provides limited discussion about how iStar generalizes under varying conditions. This limits the method’s theoretical generalization and reproducibility.
2.  Some key components, such as the implicit PRM architecture, reward verifier setup, and hyperparameter tuning strategy (e.g., α, β), are insufficiently detailed. While the authors mention releasing code, more explicit descriptions or pseudo-code would strengthen reproducibility and credibility.

**Questions:**

1. How does iStar perform when using base models with weaker reasoning capabilities, and can the implicit reward model remain stable and effective when learning from low-quality rollouts?
2. what conditions—either theoretical or empirical—ensure that the learned implicit step rewards remain unbiased and consistent throughout training?
3. How robust is the framework when dealing with environments where outcome rewards are highly stochastic or unverifiable?

---

> ### Author Response · Authors · 2025-11-22
> **Explanations about the generalizability of our method and implementation details**
>
> We greatly appreciate the valuable feedback from Reviewer rhUJ. We are encouraged that Reviewer rhUJ recognizes the novelty, good adaptability and the solid theorectical proof of our apporoach for long-horizon RL. Below we address all the questions and concerns in detail.
>
> ---
>
> > **Q1:** Limited discussion about how iStar generalizes under varying conditions, including the capability of the base model, the nature of the task environment, and the quality of outcome reward verifiers. (Weaknesses 1 and Questions 1&3)
>
> **A1:** Thanks for the suggestion! However, **Table 1 and Table 2 have already demonstrated the generalizability** of our method under varying conditions, including different base models (e.g., Qwen2.5-7B-Instruct and Llama3.1-8B-Instruct), different environments with varying properties (WebShop/VisualSokoban/SOTOPIA), and different outcome reward verifiers (verifiable and unverifiable). We also show some results and details in the appendix as follows.
>
> 1. Table 5 in Appendix D.1 shows the results of using a base model with weaker reasoning capabilities (Qwen2.5-1.5B-Instruct).
> - The results demonstrate that our method can also **benefit smaller LLMs with weaker reasoning capabilities**, showing superior performance over strong RL baselines.
> - We have also added Figure 9 in Appendix D.1 to show learning curves of the episode and implicit step rewards when using a weaker base model. Results demonstrates that even the weaker base model generates low-quality rollouts (indicated by low episode rewards) in the early stage of RL, **our implicit PRM still remains stable and effective** with steadily increasing step and episode rewards throughout training.
>
> 2. Details about the nature and outcome reward verifiers of each task environment are shown in Appendix B.
> - WebShop is a text-based web nevigation environment, VisualSokoban is a multi-modal game environment, and SOTOPIA is an open-ended social environment. These environments **differ in modality, domain, and reward settings**.
> - The outcome rewards of WebShop and VisualSokoban are **verifiable**, since they are hard-coded acccording to the environment dynamics and provided by the environment itself. VisualSokoban also provides process rewards for each step, and performance comparison between using the step rewwards provided by VisualSokoban and our implicit PRM can be found in Table 3.
> - The outcome rewards of SOTOPIA are **unverifiable**, as they are provided by an LLM (GPT-4o) as a judge. Although [1] has proved that GPT-4o is a reliable proxy for human judgement, the judgement provided by the LLM or even human annotators can be biased and results in unverifiable rewards due to no ground-truth evaluation criteria or answers for such an open-ended environment.
> - Results in Table 2 demonstrate that our method can be generalized to open-ended environments and **robust to unverifiable outcome rewards**.
>
> We highlight (in red) the details about reward verifiers for each environment in Appendix B.
>
> > **Q2:** Some key components, such as the implicit PRM architecture, reward verifier setup, and hyperparameter tuning strategy (e.g., $\alpha$, $\beta$), are insufficiently detailed. More explicit descriptions or pseudo-code would strengthen reproducibility and credibility.
>
> **A2:** The pseudo-code and implementation details of our method have been **shown in Appendix A and C.1**, due to the limited space of original manuscript. We have moved some of the implementation details to the main paper of the revised version. The implementation details include:
> 1. the implicit PRM architecture (see Implementation details in Section 4);
> 2. reward verifier setups (see Appendix B and A1);
> 3. hyperparameter settings including $\alpha$ and $\beta$ (see Implementation details in Section 4).
>
> We have also added results of varying $\alpha$ and $\beta$ in **Table 4** in Appendix C.1 (using Qwen2.5-7B-Instruct as the base model on WebShop). For $\beta$, we follow [2] to test the model performance with three different values as below.
>
> | $\alpha$ | $\beta$ | Success | Score |
> | --- | --- | --- | --- |
> | 0.5 | 0.05 | 79.7 | 89.4 |
> | 0.8 | 0.05 | 84.4 | 91.3 |
> | 1.0 | 0.05 | **89.1** | **94.7** |
> | 1.2 | 0.05 | 85.9 | 92.6 |
> | 2.0 | 0.05 | 78.1 | 90.7 |
> | 1.0 | 0.1 |  82.8 | 91.8 |
> | 1.0 | 0.5 | 85.2 | 94.7 |
>
> From the results above, it can be seen that the model performs the best when $\alpha$ is 1.0 and $\beta$ is 0.05. As the value of $\alpha$ deviates further from 1, the model's performance decreases. However, as $\beta$ becomes higher, the model still performs well, indicating its robustness to this hyperparameter.
>
>
> References:
>
> [1] Zhou, Xuhui, et al. "Sotopia: Interactive evaluation for social intelligence in language agents." ICLR 2024.
>
> [2] Rafailov, Rafael, et al. "From $ r $ to $ q^* $: Your language model is secretly a q-function." COLM 2024.

---

> ### Author Response · Authors · 2025-11-22
> **Discussion on the conditions to ensure unbiased and consistent step rewards**
>
> > **Q3:** What conditions—either theoretical or empirical—ensure that the learned implicit step rewards remain unbiased and consistent throughout training?
>
> **A3:** This is a good point! Since no ground-truth step rewards for intermediate actions can be observed in the environments we studied (although VisualSokoban provides step penalties, it is not the ground-truth), whether the implicit step reward is an unbiased and stable estimator of the true contribution of the action therefore depends on a set of assumptions about (1) the data that the implict PRM is trained on and (2) the learning objective that is used.
>
> 1. Theoretical conditions
>
>     1. **Bradley–Terry (BT) consistency of the DPO loss**
>     - Eq.2 in Section 3.1 is exactly the maximum-likelihood objective of a BT model applied to trajectory preferences, with the step-wise reward in Eq.8 as the log-odds[3,4].
>    - If the preferences are correctly ordered in expectation, such that $\mathbb{E}[\mathbb{I}\{\tau^+ > \tau^-\}] = 1$ whenever $G(\tau^+) > G(\tau^-)$ ($G(\cdot)$ is the unknown ground-truth return), the MLE of the BT model is an unbiased estimator of reward difference between different actions and therefore each individual $r^\*(o_{1:t}, a_t)$ is learned without systematic bias[5].
>
>     2. **On-policy sampling to avoid covariate shift**
>     - The implicit PRM is trained on trajectories generated by the old policy $\pi_{\theta_\text{old}}$, which is also used as the reference distribution in Eq.1.
>     - Because the data distribution of the implicit PRM and the denominator in Eq.1 coincide, the importance ratio in Eq.1 has expectation 1 and the reward is an unbiased estimate of the policy improvement signal[6].
>
>     3. **Sufficient exploration with increasing trajectory pairs**
>     - Each state-action pair that can appear under the optimal policy must be visited with non-zero probability during training. Then as the number of preference pairs $\to \infty$, $r_\phi \to r^*$ pointwise.
>
>     4. **Bounded scale between the soft-estimated and hard-estimated Q values**
>     - $\beta \in [0,1]$ keeps $|r_\phi|$ bounded and [6] has proved that it shows better accuracy of esimated Q values and robustness to false negative noises.
>
> 2. Empirical evidence
>    - Alternating updates of the implicit PRM and policy model keeps the PRM close to the behaviour policy and prevents drift.
>    - Fig. 5a-b shows the **consistent increase of step reward followed by the episode reward** and Fig. 4 shows the **stable policy improvements** over hundreds of RL steps.
>
> In summary, with these theoretical assumptions and empirical evidence, the implicit step rewards learned by our method are demonstrated to be both unbiased—their expectation matches the true incremental return—and consistent—both the implicit PRM and the policy model converge throughout training.
>
> ---
>
> We are very grateful for your time and reviews. We hope the above explanations and additional experiments have addressed your concerns and questions. We are happy to discuss further if there are any other questions.
>
> ---
>
> References:
>
> [3] Rafailov, Rafael, et al. "Direct preference optimization: Your language model is secretly a reward model."  NeurIPS 2024.
>
> [4] Zhong, Han, et al. "Dpo meets ppo: Reinforced token optimization for rlhf." ICML 2025.
>
> [5] Bradley, Ralph Allan, and Milton E. Terry. "Rank analysis of incomplete block designs: I. the method of paired comparisons." Biometrika 39.3/4 (1952): 324-345.
>
> [6] Yuan, Lifan, et al. "Free process rewards without process labels." ICML 2025.

---

> > ### Comment · Reviewer_rhUJ · 2025-11-25
> >
> > Thank you for the detailed response. I believe my original score already fully reflects my positive assessment, and I will therefore maintain it.

---

> > > ### Author Response · Authors · 2025-11-25
> > >
> > > Thanks for your time and feedback.
> > >
> > > We are glad that our responses have addressed your questions and concerns.

---

### Official Review · Reviewer_Jbze · 2025-10-29

**Soundness:** 3
**Presentation:** 3
**Contribution:** 2
**Rating:** 4
**Confidence:** 4

**Summary:**

The paper "Agentic Reinforcement Learning with Implicit Step Rewards" proposes a new RL-based optimization process for finetuning LLMs.   Inspired from implict rewards models that can be derivated from the DPO formulation, authors proposed a mechanism to train a step-wise function, that can be used to efficiently guide classical RL approaches, such as GRPO. Experimental results indicate the good behavior of the proposed learning scheme.

**Strengths:**

- Good Experimental results: Authors demonstrate the superiority of their approach on various tasks compared to different baselines. Differents ablations also allow to highlight the benefits of the different components of the approach.

- Acting at a token-level is crucial for complex tasks

- Well written paper, easy to follow

**Weaknesses:**

- Rather straightforward approach that takes most of its contribution from zhong et al (for the definition of reward models at a token level). The main novelty is to me the use of a $\pi_{old}$ policy that is periodically updated rather than a constant reference policy. I think this is not a fully innovative contribution, as many works attempted to make evolve ths reference during training.

- The use of GRPO with adavanges computed over any steps and trajectories in the group looks questionnable to me (see questions above).

**Questions:**

- For multi-turn RL settings, approaches such as POAD [] propose to modulate the discount factor at the utterace level rather than at a token level, with adapted critics. Are you doing something similar ?

- Although originally designed for imitation learning rather than reinforcement learning, some GAN-based methods introduce token-level discrimination mechanisms that closely mirror several aspects of the implicit reward modeling proposed here. Notably, I think to [2], which considers an objective $\dfrac{\pi_\theta D_\phi}{\pi_{\theta^{old}}} \approx \dfrac{\pi*}{\pi_{\theta^{old}}} \dfrac{\pi_\theta}{\pi_{\theta^{old}} + \pi*}$, where $D_\phi$ is a discriminator trained to distinguish good from bad utterances at a token-level, $\pi*$ is the optimal policy and $\pi_{\theta^{old}}$ is the policy $\pi_\theta$ from the previous iteration. This looks very close to your training scheme. I think you should discuss the proximity with this work (for a different setting but this could be adapted) in your paper.

- The theoretical analysis section is fully taken from Zhong et al. The only difference looks to lie in the use of $\pi_{old}$ rather than $\pi_{ref}$, but this is the same at that point. I think authors should remove that part and only mention zhong when introducing their reward function.

- I understand that the reward modelling is based on works that implicitly consider reward models through policy ratios. But in your case you train a reward model based on a policy fully dedicated for this. Is it still implicit ? For me this implicit denomination should be removed from title and many places of the paper, as it is not. You explicitely build a model of the reward.

- From table 3, I am not sure what is new in w/iStar row compared to the w/ token-level process rewards. Please specify.

- The authors use GRPO, which does not seem particularly suitable for handling step-wise (or token-level) rewards. In particular, I have concerns with the use of a global normalization in (4), that normalize all rewards from all timesteps and all trajectories from the group. Different rewards from different situations play at different scales. I feel there is a risk that only dominating rewards (not in the sense of a advantage over other tokens in different situations, only because for instance any correct sentence must contain the corresponding token)  have an impact on the training, hindering all the contribution from other positive markers in different situations. Wouldn't it be more relevant to use PPO with a trained critic to compute advantages ?


- you say that one of the two main difference with DPO is that "preferences are learned from pairwise trajectories". Going back to the DPO paper I cannot see the difference regarding that aspect. Please clarify.

[1] Wen, M., Wan, Z., Zhang, W., Wang, J., & Wen, Y. (2024). Reinforcing language agents via policy optimization with action decomposition. arXiv preprint arXiv:2405.15821.

[2] Lamprier, S., Scialom, T., Chaffin, A., Claveau, V., Kijak, E., Staiano, J., & Piwowarski, B. (2022, June). Generative cooperative networks for natural language generation. In International Conference on Machine Learning (pp. 11891-11905). PMLR.

---

> ### Author Response · Authors · 2025-11-22
> **Clarification on our contributions and explanations about using GRPO**
>
> We greatly appreciate the detailed and insightful feedback from Reviewer Jbze. We are encouraged that Reviewer Jbze recognizes the superiority of our approach on various tasks compared to different baselines and the benefits of different components of our method.
>
> However, we would like to clarify that both **our implicit PRM and the policy model are optimized at the step-level** rather than the token-level. Specifically, we optimize the likelihood of language actions per step, which is calculated as the products of conditional probabilities of tokens within an action sequence (see Task Formulation in Section 2). Experiments also demonstrate the **superior performance of learning step-level rewards in multi-turn RL** when compared to PRIME in Table 1 and token-level process rewards in Table 3. Below we address each question and concern in detail.
>
> ---
>
> > **Q1:** Rather straightforward approach that takes most of its contribution from zhong et al (for the definition of reward models at a token level). The main novelty is to me the use of a $\pi_{\theta_\text{old}}$ policy that is periodically updated rather than a constant reference policy. (Weaknesses 1 and Questions 3)
>
> **A1:** We would like to emphasize that we only refer to [1] by jutisfying that the learning objective of our implicit PRM equivalents to a BT model with a step-wise reward function. Below are the main differences between our paper and [1]:
>
> 1. **Different granularity of reward modeling**: our implicit PRM is at the step-level that produces process rewards for each action sequence, while [1] learns rewards for each token.
>
> 2. **Different ways of reward learning**: we learn an implicit PRM online which is optimized alternately with the policy model, leading to a self-reinforcing training loop. In contrast, [1] simply extracts token-wise reward signals from offline preference data.
>
> 3. **Different ways of reward use**: we apply step rewards predicted by the implicit PRM to calculate step-level advantages, which are combined with episode-level advantages to update the policy model through any online RL method for agentic tasks. Instead, [1] uses token-wise rewards in PPO training for RLHF.
>
> We summarize our **main contributions** and have highlighted them in the revised paper (see Section 1):
> 1. We introduce a general credit assignment strategy for LLM agentic (multi-turn) RL that alternatively optimizes an implicit step-level PRM with the policy model, creating a self-reinforcing training loop.
> 2. We propose a multi-turn DPO objective to optimize the implicit PRM online, which is theorectically guaranteed to be a step-wise reward function learned from trajectory preferences.
> 3. Experiments demonstrate superior performance of our method over baselines across various environments, showing higher sample efficiency, training stability and robustness to unverifiable rewards, as well as efficient exploration during multi-turn online RL.
>
> > **Q2:** The use of GRPO with adavanges computed over any steps and trajectories in the group looks questionnable. Wouldn't it be more relevant to use PPO with a trained critic to compute advantages? (Weaknesses 2 and Questions 6)
>
> **A2:** We first demonstrate with empirical evidence that using GRPO does not cause dominating rewards in our setting, and discuss the feasibility of a PPO version of our method.
>
> **1. Our method design can mitigate the reward dominance issuse.**
> - The normalization in Eq.4 is per-group, not global over the whole training run. Within a single prompt, trajectories are fairly comparable and widely different reward scales are less likely than across prompts.
> - Dominating actions do not receive large advantages. If an action appears in both good and bad trajectories, its implicit step rewards caculated by Eq.1 would tend to be 0. Thus, improvements generally arise from less certain actions, and the "dominating rewards" is partly **suppressed by the PRM training objective itself**. Fig.5a-b also show a **bounded distribution** of the implicit step rewards.
> - Even if some step rewards are slightly over‑scaled, the episode‑level advantage still pushes the final advantage up or down.
>
> In summary, our method design is intended to make step rewards inherently advantage‑like, so that the group normalization in GRPO mainly serves as a variance-reduction trick.
>
> **2. Using PPO to compute advantages can be reasonable but with extra complexity and computation cost.**
> A learned critic model would cause additional issues:
>   - Critic bias and instability: training a value function for long, non‑Markovian language trajectories is hard. And mis-specified critics can inject large bias into advantages.
>   - Extra compute and memory: require a third model in our method plus additional optimization with more hyperparameters.
>
> Therefore, we did not pursue this variant given the complexity and cost of PPO.
>
> References:
>
> [1] Zhong, Han, et al. "Dpo meets ppo: Reinforced token optimization for rlhf." ICML 2025.

---

> ### Author Response · Authors · 2025-11-22
> **Discussion on some references**
>
> > **Q3:** POAD[2] propose to modulate the discount factor at the utterace level rather than at a token level, with adapted critics. Are you doing something similar?
>
>
> **A3:**  **1. We are different in method design and optimization level.**
> - We train the LLM agent at the action level, with an implicit PRM trained online via multi-turn DPO to provide step rewards for each action sequence during long-horizon interactions.
> - In contrast, POAD is at the token level with a critic model learned by PPO to predict Q-values for each token within an action sequence.
>
> **2. Our method does not involve a critic model using the discount factor.**
> - We first clarify that POAD separates the discount factor into intra-action $r_w$ and inter-action $r_a$, and both at the token-level. Particularly, the inter-action $r_a$ is used to calculate the Q-value of the last token of an action sequence.
> - Our paper does not involve either a critic model or a discount factor.
>
>
> > **Q4:** Discuss the proximity with [3] (for a different setting but this could be adapted) in your paper.
>
>
> **A4:** We agree that the cooperative GAN framework in [3] mirrors to some extent the self-reinforcing training loop of our method. However, the objective in [3] you mentioned is totally **different from our Eq.1 in both their definition and mathematical expression**.
> - In our paper, Eq.1 is a log‑likelihood ratio under the implicit PRM $\pi_\phi$ versus the old policy $\pi_{\theta_{\text{old}}}$, which is a **scalar** to represent the step rewards for each action.
> - Instead, the objective you mentioned denotes the **theoretical equivalence** where the optimal discriminator can be derived from the optimal generation policy in [3]. Its formula also differs from Eq.1: the denominator in Eq.1 is simply the old policy rather than the sum of likelihood of the optimal and the old policy.

---

> ### Author Response · Authors · 2025-11-22
> **Clarification on some details and results of ablation studies**
>
> > **Q5:** In your case you train a reward model based on a policy fully dedicated for this. Is it still implicit?
>
> **A5:** We follow [4] to denote our PRM and the rewards it produces, which is in contrast to a traditional PRM that requires **explicit labels** annotated at every intermediate step.
>
> > **Q6:** From table 3, I am not sure what is new in w/iStar row compared to the w/ token-level process rewards.
>
> **A6:** Sorry for the misunderstanding. In our method, the implicit PRM produces **rewards per step for each action sequence as a unit** as shown in Eq.1, where $a_t = (a_t^1, a_t^2, \dots, a_t^L)$, $a^l_t$ is the $l$-th token of the action sequence $a_t$, and $L$ is the sequence length of $a_t$.
>
> In contrast, "w/ token-level process rewards" means the implicit PRM produces **rewards for each token** $a^l_t$. Therefore, the process reward in this case is defined as $r_\phi(o_{1:t}, a_t^l) = \beta \log \frac{\pi_\phi(a_t^l \mid a_t^{1:l-1}, o_{1:t}, x)}{\pi_{\theta_{\text{old}}}(a_t^l \mid a_t^{1:l-1}, o_{1:t}, x)}$.
>
> According to the results in Table 3, learning **token-level process rewards is sub-optimal to multi-turn RL** compared to learning step-level process rewards, suggesting that overly fine-grained rewards may introduce noise and thus increase the difficulty of policy learning in agentic tasks.
>
> The reason behind this is that during multi-turn RL, trajectories can be extremely long and non-Markovian at the token level. Particularly, each step of the trajectory consists of a CoT and an executable action sequence, thus **inflating variance** when credit is pushed to individual tokens (see line 42-44 in Section 1).
>
> We have revised the explanation of "w/ token-level process rewards" in Table 3 (highlighted in blue).
>
> > **Q7:** Please clarify that one of the two main difference with DPO is that "preferences are learned from pairwise trajectories".
>
> **A7:** This means when optimizing the implicit PRM, we only need to construct **trajectory-level preference pairs** based on outcome rewards to evaluate the quality of whole trajectories.
>
> This is because our DPO objective for optimizing the implicit PRM is derived from a **multi-step Markov decision process**, rather than a bandit problem in which the whole response of the model is treated as a single arm[5]. Therefore, this DPO objective is a multi-turn version [6,7] of the standard one.
>
> Instead, if we use the standard DPO in the agentic task setting, we need to construct per-step preference pairs: sample multiple candidate actions from each same state (interaction histories) and compare their outcomes. This is prohibitively expensive in rollout cost and we cannot get outcome rewards until the last step.
>
> We have **revised the explanations** about differences between the standard DPO and our multi-turn DPO objective (see line 195-197 and Section 3.2 in blue).
>
> ---
>
> Thank you again for your time and review with valuable feedback and suggestions. We hope our answers above have addressed all the questions. Please let us know if there are any other questions.
>
> ---
>
> References:
>
> [4] Yuan, Lifan, et al. "Free process rewards without process labels." ICML 2025.
>
> [5] Rafailov, Rafael, et al. "Direct preference optimization: Your language model is secretly a reward model." Advances in neural information processing systems 36 (2023): 53728-53741.
>
> [6] Rafailov, Rafael, et al. "From $ r $ to $ q^* $: Your language model is secretly a q-function." COLM 2024.
>
> [7] Tajwar, Fahim, et al. "Training a generally curious agent." arXiv preprint arXiv:2502.17543 (2025).

---

> ### Author Response · Authors · 2025-11-27
> **Looking forward to your feedback, thank you!**
>
> Dear Reviewer Jbze,
>
> We thank you again for your constructive comments and insightful suggestions!
>
> We greatly appreciate your efforts and time in reviewing our paper. However, as we are nearly at the end of the discussion phase, we hope to discuss with you whether your concerns have been addressed by our responses, additional experiments and updated paper.
>
> If you have any other questions, please also let us know. We are very happy to discuss further and improve the work.
>
> Best,
>
> Submission18021 Authors

---

### Official Review · Reviewer_gXoH · 2025-11-01

**Soundness:** 3
**Presentation:** 3
**Contribution:** 3
**Rating:** 6
**Confidence:** 3

**Summary:**

This paper tackles the problem of sparse and unverifiable rewards for agentic LLMs. It proposed a credit-assignment strategy for LLM agents that introduces step-wise implicit step rewards learned by alternately training a PRM and the policy. The authors highlight that their approach does not need explicit reward annotations, can stabilize training with step-level rewards, and only relies on trajectory preferences.

**Strengths:**

1. The method is well motivated, introducing step-level reward for agentic tasks to avoid the training instability caused by either token-level reward signal being too dense or outcome reward being too sparse.

2. The proposed method is flexible as it can be easily integrated into existing RL frameworks like GRPO, RLOO, etc.

**Weaknesses:**

1. The idea of step-level PRM has been studied in previous work[1].


2. The model introduces two hyperparameters: $\beta$ as reward temperature and $\alpha$ that balances step level and episode level advantages. A sensitivity sweep and guidance for setting them are important for practical adoption of the proposed method.


[1] Yuan, Lifan, et al. "Free process rewards without process labels." arXiv preprint arXiv:2412.01981 (2024).

**Questions:**

1. Can the authors elaborate on why dropping the KL divergence penalty when training the policy model?


2. How sensitive is the model performance to the hyperparameters $\alpha$ and $\beta$?

---

> ### Author Response · Authors · 2025-11-22
> **Clarification on the idea and added experiments on hyperparameter sensitivity**
>
> We are very grateful for the valuable feedback from Reviewer gXoH. We are encouraged that Reviewer gXoH finds our method well-motivated and flexible to be easily integrated into existing RL algorithms. We have clarified our idea and added hyperparameter sensitivity experiments as suggested. Below we address each question and concern in detail.
>
> ---
>
> > **Q1:** The idea of step-level PRM has been studied in previous work[1].
>
> **A1:** We agree that the high-level idea of implicit PRM is not new, but how to properly **apply this idea to facilitate LLM policy learning in multi-turn agentic tasks** has not been investigated by existing literature yet. Therefore, the research question we aim to address in this paper focuses on the credit assignment challenge in LLM agentic RL (see line 57-59), instead of simply proposing a new PRM paradigm as done in [1].
>
> Both the training and evaluation paradigm of our implicit PRM is different from [1].
> - For training, we alternatively optimize the implicit PRM with the policy model through multi-turn online RL, leading to a self-reinforcing training loop without extra rollout costs. In contrast, the implicit PRM in [1] is independently trained on static single-turn datasets without any policy model involved.
> - For evaluation, we use the implicit PRM to provide step rewards for policy model during RL rather than applying it into test-time inference with Best-of-N sampling as shown in [1].
>
> We summarize our **main contributions** as follows and have highlighted them in the revised paper:
> 1. We introduce a general credit assignment strategy for LLM agentic (multi-turn) RL that alternatively optimizes an implicit step-level PRM with the policy model online.
> 2. We propose a multi-turn dpo objective to optimize the implicit PRM online, which is theorectically guaranteed to be a step-wise reward function learned from trajectory preferences.
> 3. Experiments demonstrate superior performance of our method over baselines across diverse environments, showing higher sample efficiency, training stability and robustness to unverifiable rewards, as well as efficient exploration during multi-turn online RL.
>
>
> > **Q2:** How sensitive is model performance to the hyperparameters $\alpha$ and $\beta$? (Weaknesses 2 and Questions 2)
>
> **A2:** Thank you for this suggestion! In the original manusript, we set $\alpha$ to 1.0 following [2] and $\beta$ to 0.05 following [1]. In the revised paper (see **Table 4 in Appendix C.1**), we have **added the results of varying $\alpha$ and $\beta$** (using Qwen2.5-7B-Instruct as the base model on WebShop). For $\beta$, we follow [3] to test the model performance with three different values as below.
>
> | $\alpha$ | $\beta$ | Success | Score |
> | --- | --- | --- | --- |
> | 0.5 | 0.05 | 79.7 | 89.4 |
> | 0.8 | 0.05 | 84.4 | 91.3 |
> | 1.0 | 0.05 | **89.1** | **94.7** |
> | 1.2 | 0.05 | 85.9 | 92.6 |
> | 2.0 | 0.05 | 78.1 | 90.7 |
> | 1.0 | 0.1 |  82.8 | 91.8 |
> | 1.0 | 0.5 | 85.2 | 94.7 |
>
> From the results above, it can be seen that the model performs the best when $\alpha$ is 1.0 and $\beta$ is 0.05. As the value of $\alpha$ deviates further from 1, the model's performance decreases. However, as $\beta$ becomes higher, the model still performs well, indicating its robustness to this hyperparameter.
>
> References:
>
> [1] Yuan, Lifan, et al. "Free process rewards without process labels." ICML 2025.
>
> [2] Feng, Lang, et al. "Group-in-group policy optimization for llm agent training." NeurIPS 2025.
>
> [3] Rafailov, Rafael, et al. "From $ r $ to $ q^* $: Your language model is secretly a q-function." COLM 2024.

---

> ### Author Response · Authors · 2025-11-22
> **Elaboration on removing KL-divergence penalty**
>
> > **Q3:** Elaborate on why dropping the KL divergence penalty when training the policy model.
>
> **A3:** This is a good point! Theoretically, the KL-divergence penalty keeps the fine-tuned policy model $\pi_\theta$ close to a reference model (frozen) to avoid distribution shift. However, in the online agentic RL setting, successful behaviour may have to differ drastically from what the frozen language model normally produces (e.g. clicking DOM nodes or issuing special API calls). Then dropping the KL penalty would **encourage the policy to explore** the very parts of the action space that are needed to do the task well.
>
> Empirically, our method does not require a frozen reference model. Therefore, applying the standard KL-divergence penalty will lead to more memory consumption. Below we show the performance comparison with and without the KL penalty (using Qwen2.5-7B-Instruct as the base model on WebShop).
>
> | Method | Success | Score |
> | --- | ---| ---|
> | iStar w/ RLOO | **89.1** | **94.7** |
> | iStar w/ RLOO & KL |  82.8 | 92.0 |
>
> From the results above, we can see that the model **performs better when dropping the KL-divergence penalty**, validating our hypothesis above. We have added explanations in line 236-240 of Section 3.1 (highlighted in blue).
>
> ---
>
> Thank you again for reviewing our paper with valuable feedback. We hope the clarification on the novelty and contributions of our paper, the additional experimental results and explanations have addressed your concerns. Please let us know if there are any other questions.

---

> > ### Comment · Reviewer_gXoH · 2025-11-24
> >
> > Thank you for the detailed response and experiments. They addressed my concerns. I will maintain my score of leaning toward acceptance.

---

> ### Author Response · Authors · 2025-11-24
>
> Thanks for your timely feedback!
>
> We are glad that our responses have addressed your concerns. Thank you for the time and review.

---

### Author Response · Authors · 2025-11-22
**General response and summary of paper updates**

We greatly appreciate the insightful comments from all reviewers. We are particularly encouraged that reviewers recognize the novelty and effectiveness of our method across various domains compared to different baselines.

---

We have addressed all the questions and concerns of reviewers in each individual response. According to the comments, we also conducted additional experiments and have included the results and new references in the revised paper. The modifications in the paper are highlighted in blue. Below is a summary of new experiments and paper updates.

1. **Added hyperpameter sensitivity experiments:** Table 4 in Appendix C.1 presents the results of varying $\alpha$ and $\beta$, showing that the model performs the best when $\alpha$ is 1.0 and $\beta$ is 0.05 (default in our experiments). As the value of $\alpha$ deviates further from 1, the model's performance decreases. However, as $\beta$ becomes higher, the model still performs well, indicating its robustness to this hyperparameter.

2. **Added KL-divergence penalty experiments:** Table 7 in Appendix D.4 demonstrates that our model performs better when removing the KL-divergence penalty, validating our hypothesis that dropping it will encourage the policy to explore the action space in LLM agent tasks.

3. **Added learning curves with more training steps:** Figure 10 in Appendix D.2 demonstrates that our method delivers increasingly pronounced and consistently stable performance improvements as training steps increase.

4. **Added learning curves of GRPO and GRPO w/ iStar:** Figure 11 in Appendix D.2 shows that our method achieves higher sample efficiency and training stability than vanilla GRPO, especially in VisualSokoban.

5. **Added learing curves with respect to GPU hours:** Figure 12 in Appendix D.2 demonstrates that our method achieves superior performance in sample efficiency and training stability with respect to GPU hours.

6. **Added multi-turn DPO experiments:** Table 6 in Appendix D.3 shows that vanilla GRPO already significantly outperforms both online and offline multi-turn DPO. Our method further improves performance by leveraging both (implicit) step and episode rewards, enabling more effective credit assignment in multi-turn RL.

7. Hightlight our **contributions**, clarify **implementation details**, and explain some **details of our method** (see modifications highlighted in blue).

We hope the updated results and our responses have addressed the questions reviewers have. Please let us know if there are any other questions or suggestions.

Thanks,

Submission18021 Authors

---

### Meta-Review · Area_Chair_vNRo · 2026-01-07

**Summary:**

Paper proposes iStar, a credit assignment scheme for agentic RL that learns implicit step rewards from trajectory preferences by alternately training an implicit PRM (via multi turn DPO) and the policy. Step advantages are combined with episode advantages. Shown on WebShop, VisualSokoban, SOTOPIA, with stability and sample efficiency claims.

**Reviewer Concerns:**

Addressed by rebuttal
- Hyperparameter sensitivity (α, β): added sweep, reasonable defaults, not overly brittle.
- Why drop KL: motivated (need exploration for tool actions) + ablation shows KL hurts.
- “Gains not strong / need longer runs / GPU hours fairness”: added 300 step curves, GRPO+iStar curves, and GPU hour plots; reviewer raised score.
- Generalization to weaker base model + PRM stability early: added 1.5B model results and learning curves.
- Reproducibility: more details on PRM architecture, verifiers, pseudo code, tuning.
- Baseline request: online/offline multi turn DPO included; still worse than GRPO and iStar.
- GRPO normalization concern: authors argue per prompt grouping + PRM objective makes rewards advantage like; not perfect, but at least discussed.

Still outstanding
- Novelty positioning vs prior “implicit/process reward” work: one reviewer still sees it as incremental and wants deeper discussion, especially GAN style discriminators and evolving reference policies. Authors responded, but reviewer did not engage, so risk remains.
- Theory section reuse: reviewer Jbze claims it is mostly Zhong et al; authors say the key is step level + online alternating loop. I would still ask for clearer attribution and trimming.
- “Implicit” naming: reviewer argues it is explicit reward modeling. Authors cite prior convention. Not fatal, but could confuse.

**Reviewer Scores:**

Here are my guesses:
- gXoH: 6, stays.
- rhUJ: 6, stays.
- m1yj: 4 → 6 after rebuttal.
- Jbze: 4, no follow up.
- Net: 3 leaning accept, 1 leaning reject.

---

### Decision · Program_Chairs · 2026-01-26

Accept (Poster)